# OneCat: Decoder-Only Auto-Regressive Model for Unified Understanding and Generation

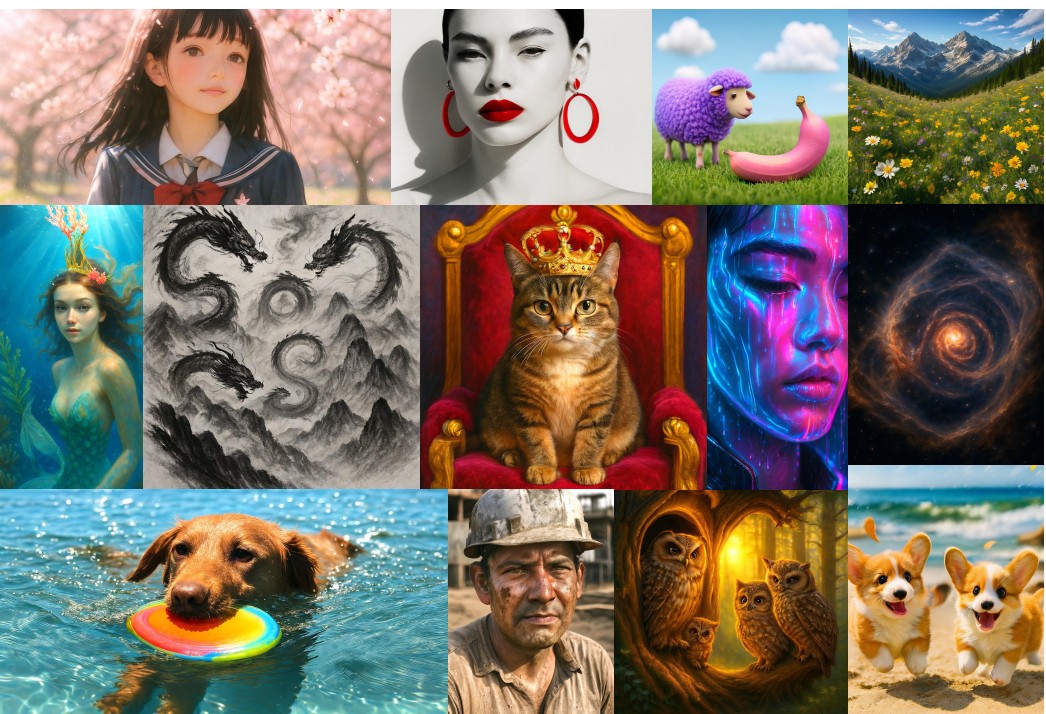

Figure 1: Showcases of the text-to-image generation abilities of the **OneCAT** model.

## Abstract

We introduce OneCAT, a unified multimodal model that seamlessly integrates understanding, generation, and editing within a single decoder-only transformer architecture. OneCAT uniquely eliminates the need for external components such as Vision Transformers (ViT) or vision tokenizer during inference, leading to significant efficiency gains, especially for high-resolution image inputs and outputs. This is achieved through a modality-specific Mixture-of-Experts (MoE) design trained with a unified autoregressive (AR) objective, which also natively supports dynamic resolutions. Furthermore, we pioneer to achieve multi-scale visual autoregressive mechanism within the Large Language Model (LLM) with proposed scale-aware adapter (SAA) that drastically reduces decoding latency compared to diffusion-based methods while maintaining state-of-the-art performance. Our findings demonstrate the powerful potential of pure autoregressive modeling as a sufficient and elegant foundation for unified multimodal intelligence. As a result, OneCAT outperforms existing unified models across benchmarks for multimodal understanding, generation, and editing.

## 1 Introduction

Modular approaches using separate modules for understanding (Wang et al., 2024a; Bai et al., 2025; Chen et al., 2024c), generation (Shi et al., 2020; Labs, 2024; Esser et al., 2024),

and editing tasks (Zhang et al., 2023; Brooks et al., 2023; Zhang et al., 2025) are becoming dominant for multi-modal frameworks. Despite producing capable systems, such designs inherently create complex multi-stage pipelines suffering from architectural bottlenecks that limit deep, early-stage fusion of cross-modal information and introduce significant inference latency, presenting a major barrier to both efficiency and performance. Unified multimodal LLMs aim to address these limitations by integrating these disparate abilities within a single end-to-end architecture (Wang et al., 2024b; Chen et al., 2025c; Deng et al., 2025), but many methods remain tethered to the modular paradigm (Pan et al., 2025; Chen et al., 2025a). This motivates us to achieve a true revolution on the fundamental architecture that unlocks the full potential of unified systems and eschews heavy external components. In this paper, we propose a single decoder-only autoregressive model trained under a unified objective to provide elegant and potent foundation for general-purpose multimodal intelligence.

**Unified architecture.** We propose the first encoder-free framework for unified multimodal LLM (MLLM), where raw visual inputs are directly tokenized into patch embeddings and are processed alongside text tokens within a single decoder model. The critical innovation is a modality-specific MoE layer that dynamically routes continuous vision tokens, discrete vision tokens, and text tokens to specialized experts. It enables deep and early-stage feature fusion without requiring exquisite encoders (*i.e.*, ViT and vision tokenizer) for efficient inference. For generative tasks, we pioneer to embed the multi-scale autoregressive mechanism (Tian et al., 2024) into the LLM and propose a scale-aware adapter (SAA) to extract scale-specific representation for augmentation. Thus, image tokens can be predicted from low to high resolutions, while next-token prediction is adopted for text tokens. This design significantly enhances the speed and quality of image generation by simultaneously circumventing the high latency of diffusion models and learning a coarse-to-fine generative process.

**Unified training paradigm.** Training encoder-free MLLMs for strong visual perception ability is notoriously data-intensive (Luo et al., 2025). To mitigate this, existing methods like VoRA (Wang et al., 2025a) and EvE (Diao et al., 2024) perform distillation to align the internal hidden states of an LLM student with a pre-trained ViT teacher. However, they suffer from supervisory bottleneck problem that the expressive capacity of an LLM is restricted by a small teacher. We address this problem with a novel distillation strategy that first customizes a powerful MLLM teacher and then efficiently transfers its comprehensive visual perception to the proposed encoder-free unified MLLM. On such basis, we use large-scale heterogeneous multimodal data and employ a unified expert pretraining, mid-training, and supervised fine-tuning to force the shared decoder to achieve generalized representation that can seamlessly switch between comprehension, generation, and editing tasks.

Building upon these innovations, we present Only DeCoder Auto-regressive Transformer (OneCAT), a unified multimodal model. Comprehensive evaluations demonstrate that OneCAT sets a new state-of-the-art for unified models. More importantly, the encoder-free design provides a significant inference speedup, particularly for high-resolution inputs and outputs. OneCAT demonstrates the viability and superiority of a pure decoder architecture, and offers a more first-principle-aligned paradigm for multimodal modeling. It facilitates earlier cross-modal fusion through its unified MoE structure and enhances semantic consistency via unified AR objective to provide valuable insights and a powerful new baseline for developments of next-generation unified multimodal systems.

We present the related works and the differences from existing methods in Appendix A.

## 2 ONECAT

As depicted in Fig. 2, OneCAT employs a pure decoder-only architecture and eliminates the need for additional vision encoder or tokenizer **during inference**. This streamlined design significantly simplifies the model structure and reduces computational cost. Unlike existing unified MLLMs (Chen et al., 2025c; Deng et al., 2025) using semantic encoders like ViTs for understanding, we follow recent encoder-free MLLMs (Luo et al., 2025) and use a lightweight `Patch Embedding` layer to convert raw images into continuous visual tokens for efficient and lossless processing. The same `Patch Embedding` layer is also used to process

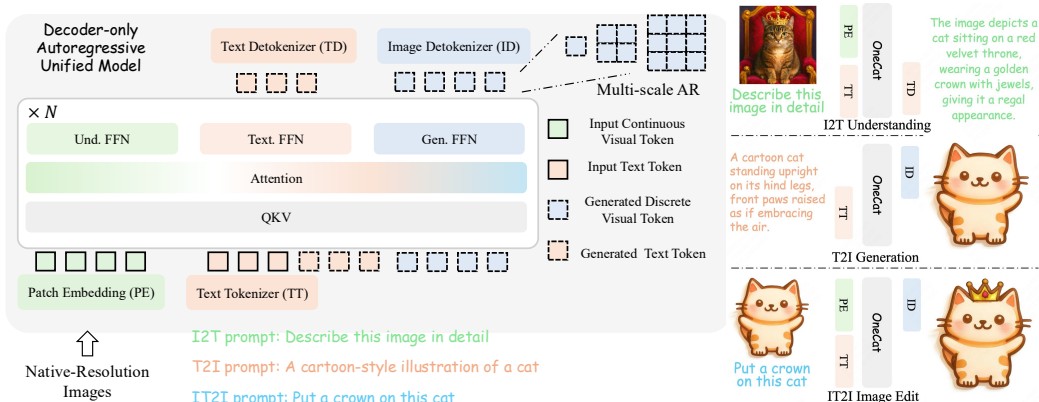

Figure 2: Inference pipeline of OneCAT, a decoder-only autoregressive unified model that seamlessly supports multimodal understanding, image generation and image editing.

reference images for editing tasks, thereby superseding separate VAE tokenizer used (Deng et al., 2025; Liao et al., 2025a) and further enhancing inference efficiency.

For text generation, OneCAT adheres to the **Next-Token Prediction** paradigm. For visual generation, it innovatively employs the **Next-Scale Prediction** (Tian et al., 2024), where images are generated in a hierarchical coarse-to-fine manner to progressively from the lowest to the highest resolution scales. Refer to Appendix B for more preliminaries.

## 2.1 MODALITY-MOE

OneCAT proposes a Modality-MoE architecture with three specialized feed-forward network (FFN) experts: a `Text. FFN` designed for text tokens for language comprehension, a `Und. FFN` designed for continuous visual tokens for visual understanding, and a `Gen. FFN` for discrete visual tokens for image generation. We employ a hard routing mechanism that assigns tokens to a specific expert based on their modality and the task at hand. Other QKV and attention layers are shared to promote parameter efficiency and robust cross-modal alignment for instruction-following.

OneCAT is initialized from the pre-trained Qwen2.5 LLM (Yang et al., 2024) to exploit its strong ability in language modeling. To construct the proposed Modality-MoE, we replicate the FFN layer from each Qwen2.5 transformer block to form three distinct experts. The core functionality for each multimodal task is handled as below.

**Multimodal Understanding.** We employ a simple yet effective patch embedding layer to convert raw images into continuous visual tokens. This layer consists of a 14×14 convolution for image patchifying, a 2×2 pixel unshuffle layer, and a two-layer Multilayer Perceptron (MLP) for projecting the visual features to match the LLM's hidden state dimension.

**Text-to-Image Generation.** We leverage a pre-trained multi-scale VAE model from Infinity (Han et al., 2025b) to map images between the pixel and latent spaces. This VAE operates with a downsampling ratio of 16 and a latent channel size of 32, and uses a bitwise quantizer (Zhao et al., 2024c) to enlarge the vocabulary. During training, the VAE tokenizer encodes target images into multi-scale discrete visual tokens to serve as ground-truth. During inference, this tokenizer is **not required**, and only the detokenizer is used to reconstruct the image from generated multi-scale visual tokens.

**Image Editing.** OneCAT seamlessly supports image editing task by leveraging a reference image and instruction input. The reference image is processed by the patch embedding layer, and the resulting continuous visual tokens are also routed to the `Und. FFN` to serve as the visual condition. The patch embedding layer provides a **near-lossless representation** of the reference image, and allows the LLM's shallower layers to obtain low-level features for pixel-level consistency, while the deeper layers to extract high-level features for semantical comprehension. Guided by this hierarchical visual context, LLM predicts new discrete visual tokens autoregressively. This design enables powerful conditional generation without any architectural modifications, showing the versatility of our unified decoder-only design.

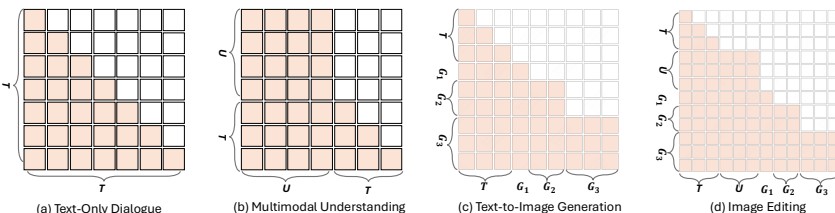

Figure 3: Multimodal versatile attention mechanism. $T$ denotes the text tokens. $U$ denotes the continuous visual tokens for multimodal understanding or reference image tokens for image editing. $G_i$ denotes the $i$-th scale discrete visual tokens for visual generation.

## 2.2 Scale-Aware Adapter for Hierarchical Generation

The tokens produced by the multi-scale VAE are inherently hierarchical (as shown in Appendix E). Specifically, lower-scale tokens mainly encode low-frequency global information such as color, illumination, and coarse structure, while higher-scale tokens capture high-frequency details including fine textures and intricate patterns. Processing these functionally divergent tokens equally with a shared `Gen. FFN` layer is not optimal and could limit the representation capacity.

To address this, we introduce the **Scale-Aware Adapter (SAA)**, a novel architectural component integrated with the `Gen. FFN`. The SAA comprises a set of parallel modules that serve as skip connections over each linear layer of the `Gen. FFN`. Each SAA is dedicated to processing tokens from a specific scale, with the total number of SAA of a `Gen. FFN` matching the number of VAE scales. During inference, discrete visual tokens are routed to corresponding scale-specific adapters based on scale indices. To ensure parameter efficiency, each adapter is achieved using a low-rank decomposition (rank $r = 64$), inspired by LoRA (Hu et al., 2022). However, unlike LoRA that is typically used for fine-tuning, the SAA modules are jointly trained in an end-to-end manner as permanent components of the LLM.

## 2.3 Multimodal versatile attention mechanism

We employ a multimodal versatile attention mechanism based on FlexAttention (PyTorch Team, 2024) to enable flexible processing of diverse modalities and tasks within a single LLM. As shown in Fig. 3, text tokens $T$ use causal attention for autoregressive generation, while continuous visual tokens $U$ apply full attention for global interaction. Multiscale discrete visual tokens $G$ use block causal attention. Tokens within the same scale attend to each other freely, while attention across scales follows a causal attention.

# 3 Model Training Pipeline

As shown in Fig. 4 and Table 1, we employ a three-stage training pipeline for OneCAT.

## 3.1 Stage-1: Multilmodal Pretraining

This stage aims to to equip OneCAT with foundational visual perception and generation abilities, while preserving its pretrained linguistic capabilities. The core challenge is that, for visual perception, the `Und. FFN` is initialized from the LLM's text-centric FFN. Although this `warm start` facilitates abstract knowledge transfer, it inherently lacks pretrained visual prior and makes the training process highly data-intensive (Luo et al., 2025). To overcome this, we leverage a custom MLLM teacher and propose a novel understanding distillation strategy to optimize `Und. FFN` that significantly enhances visual learning efficiency. In parallel, we perform generative pretraining to optimize the `Gen. FFN` and extend the LLM's next-token prediction ability to multiscale autoregressive image generation.

### 3.1.1 Stage 1-1: Teacher Training

Instead of using an off-the-shelf MLLM as teacher (*e.g.*, Qwen2.5-VL (Bai et al., 2025)), we customize a teacher to ensure parameter consistency between the LLM backbones of the

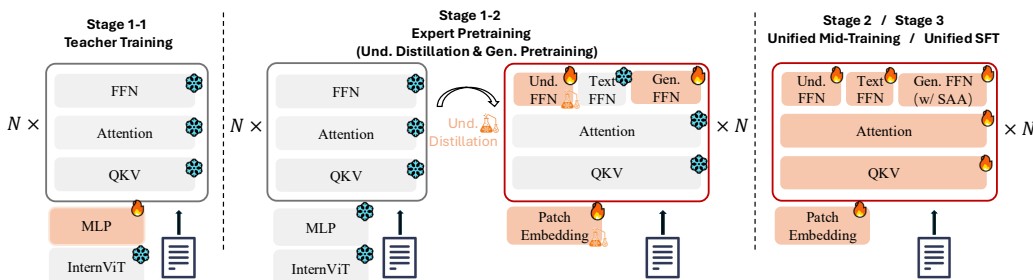

Figure 4: Training pipeline. The VAE component for visual generation is omitted for clarity.

Table 1: Detailed hyperparameter and configuration of the training recipe across stages

| Hyperparameter / Config | Stage 1-1 (Teacher Training) | Stage 1-2 (Expert Pretraining) | Stage 2 (Unified Mid-Training) | Stage 3 (Unified SFT) |
|---|---|---|---|---|
| Learning Rate | $2 \times 10^{-3}$ | $2 \times 10^{-4}$ | $2 \times 10^{-5}$ | $1 \times 10^{-5}$ |
| LR Scheduler | Cosine | Cosine | Cosine | Cosine |
| Weight Decay | 0 | 0 | 0.01 | 0.01 |
| Gradient Norm Clip | 1.0 | 1.0 | 1.0 | 1.0 |
| Batch Size | 512 | 2048 | 512 | 256 |
| Sequence Length | 1024 | 1024 | 8192 | 16384 |
| Number of Sample: Text-Only | - | - | 40M | 2M |
| Number of Sample: Und. | 10M | 436M | 70M | 11M |
| Number of Sample: Gen. | - | 52M | 60M | 3M |
| Number of Token (Total) | 5B | 0.3T | 0.6T | 57B |
| Token Ratio (T:U:G): | 0:1:0 | 0:8:1 | 1:2:6 | 1:5:6 |
| Resolution: Und. | 448×448 | 448×448 | Native | Native |
| Use thumbnail | × | × | ✓ | ✓ |
| Resolution: Gen. | - | 256×256 | Dynamical (#sides: 288∼864) | Dynamical (#sides: 288∼1776) |
| Number of Scales : Gen. | - | 7 | 10 | 10∼13 |

teacher and student, and thus, improve distillation stability and efficiency (refer to Fig. 5). Specifically, the teacher is built by connecting a pre-trained ViT (`InternViT` (Chen et al., 2024e)) and a LLM (`Qwen2.5` (Yang et al., 2024)) with a two-layer `MLP`. We freeze both ViT and LLM and only train the MLP connector on a small-scale image-text caption dataset with NTP loss. This alignment training endows the custom MLLM teacher with strong visual perception ability.

### 3.1.2 STAGE 1-2: EXPERT PRETRAINING

We then train OneCAT based on the teacher model. We freeze the `QKV`, `Attention`, and `Text FFN`, and optimize the task-specific modules: the `Und. FFN` and `Patch Embedding` layer for visual understanding, and the `Gen. FFN` for visual generation.

**Understanding Distillation**: We optimize the `Und. FFN` on a large-scale dataset of image-to-text pairs. The training objective is a combination of the NTP loss ($\mathcal{L}_{\text{NTP}}$) and a distillation loss ($\mathcal{L}_{\text{Distill}}$). Specifically, $\mathcal{L}_{\text{NTP}}$ is the cross-entropy loss for text token generation. For distillation, instead of matching the final output logits, we align the student's internal hidden states with those of the teacher model through deep feature-level matching across each transformer layer. This enables the student to not only mimic the teacher's final text prediction but also its internal computational patterns across all token (including both visual and text tokens) for better visual knowledge transfer. The distillation loss is formulated as $\mathcal{L}_{\text{Distill}} = \sum_{n=1}^{N} \text{MSE}(\mathbf{h}_S^{(n)}, \mathbf{h}_T^{(n)})$, where $\mathbf{h}_S^{(n)}$ and $\mathbf{h}_T^{(n)}$ represent the hidden state outputs from the $n$-th transformer block of the student and teacher models, respectively. The final objective is $\mathcal{L}_{\text{Und}} = \mathcal{L}_{\text{NTP}} + \lambda \mathcal{L}_{\text{Distill}}$, where $\lambda = 0.02$ is a balancing hyperparameter. Throughout distillation, all models process images at a fixed resolution of $448 \times 448$ to balance computational load and the granularity of visual features.

**Generation Pretraining**: In parallel, we optimize the `Gen. FFN` on a delicate text-to-image generation dataset to enable the LLM to learn the spatial relationships and cross-scale dependencies. We also adopt the cross-entropy loss for next-scale prediction (Tian et al., 2024) and the output image resolution is fixed to 256×256.

Table 2: Model configurations for the two variants of OneCAT. $^\dagger$ A params indicates the activated parameters and T params indicates the total parameters.

|  | Model Recipe | | | Understanding Distillation | |
| --- | --- | --- | --- | --- | --- |
|  | Base Model | A Params$^\dagger$ | T Params$^\dagger$ | Teacher ViT | Teacher LLM |
| OneCAT-1.5B | Qwen2.5-1.5B-instruct | 1.5B | 4.5B | InternViT-300M | Qwen2.5-1.5B-instruct |
| OneCAT-3B | Qwen2.5-3B-instruct | 3B | 9B | InternViT-300M | Qwen2.5-3B-instruct |

## 3.2 STAGE-2: UNIFIED MID-TRAINING

In Stage-2, we unfreeze the entire model to perform unified mid-training across multiple tasks (*i.e.,* image-to-text, text-to-image, image editing, and text-only dialogues). We incorporate the proposed `scale-aware adapter` in this stage, which is optimized with other modules together to extract scale-specific representation for enhanced image generation quality. We also introduce native resolution strategy in this stage. For visual understanding, the model is trained to process images at their original resolutions, thereby preserving fine-grained details and reducing information loss. Additionally, a thumbnail of resolution 448×448 is included to provide global visual context. For visual generation, we train the model with dynamical resolution and aspect ratios where the side lengths sampled from a range of 288 to 864 pixels, which enhancing its generation versatility and real-world applicability.

## 3.3 STAGE-3: UNIFIED SUPERVISED FINE-TUNING

The final stage involves unified supervised fine-tuning (SFT) using a curated dateset of higher-quality data to enhance instruction-following and visual generation quality. The native resolution strategy was continued, with the size of generated image expanded to support side lengths between 288 to 1776 pixels, enabling higher-resolution results.

## 4 EXPERIMENTS

### 4.1 IMPLEMENTATION DETAILS

**Datasets.** The overall curated data comprises 519 million multimodal understanding samples, 63 million visual generation samples, and 40 million text-only samples. During training, partial visual generation samples was reused, resulting in a overall training token ratio of 1:6:7 across text-only, understanding and generation. For detailed data compositions and sources at each training stage, refer to Table 1 and Appendix H.

**Model Configurations.** We conduct experiments on two model variants, OneCAT-1.5B and OneCAT-3B, which is initialized from Qwen2.5-1.5B-instruct and Qwen2.5-3B-instruct, respectively. The total parameters count of OneCAT-3B is 9B, but the activated parameters count for each token is 3B during forward process. Refer to Table 2 for more details.

**Classifier-free guidance (CFG).** We follow previous works (Chen et al., 2025b; Deng et al., 2025) to use CFG for enhanced visual generation quality. For training, we randomly drop conditional text and reference image tokens. For inference, we combines conditional and unconditional predicted logits. For more details, please refer to Appendix C.

### 4.2 PERFORMANCE EVALUATIONS

**Multimodal Understanding.** We evaluate OneCAT on several multimodal understanding benchmarks: MMBench, MME, MMMU, MM-Vet, SEED, and MathVista assess general multimodal perception and reasoning, while TextVQA, ChartQA, InfoVQA, DocVQA, GQA, and AI2D evaluate visual question answering. Table 3 compares OneCAT with three types of models: encoder-based understanding-only models, encoder-free understanding-only models, and unified MLLMs. Our OneCAT-3B model demonstrates superior performance, significantly outperforming most existing encoder-free understanding-only MLLMs, *e.g.,* HoVLE and EvEv2. For instance, on text-centric benchmarks including AI2D (77.8), ChartQA (81.2), InfoVQA (64.8), and DocVQA (91.2), OneCAT-3B achieves new state-of-the-art results among encoder-free models. It also excels in general multimodal benchmarks such as MME-P (1630), MMBench (78.8), and MM-Vet (52.2).

Table 3: Evaluation on multimodal understanding benchmarks. Benchmark details and the computation of average scores are provided in Appendix G. **A-LLM** denotes the activated LLM parameters, while **Vis.** refers to the vision encoder/tokenizer parameters. A '/' indicates models that do not require external vision component for understanding. Higher scores are better. Best in **bold**, second best is underlined (among unified models).

| Model | # Params | | VQA Benchmarks | | | | | | | General multimodal Benchmarks | | | | | | | |
|---|---|---|---|---|---|---|---|---|---|---|---|---|---|---|---|---|---|
| | A-LLM | Vis. | TextVQA | ChartQA | InfoVQA | DocVQA | GQA | AI2D | Avg. | MME-P | MME-S | MMB | MMMU | MMVet | MathVista | SEED | Avg. |
| *Encoder-based Understanding Only Models* | | | | | | | | | | | | | | | | | |
| InternVL2-2B (Chen et al., 2024d) | 1.8B | 0.3B | 73.4 | 76.2 | 58.9 | 86.9 | - | 74.1 | - | 1440 | 1877 | 73.2 | 34.3 | 44.6 | 46.4 | 71.6 | 56.2 |
| InternVL2.5-2B (Chen et al., 2024c) | 1.8B | 0.3B | 74.3 | 79.2 | 60.9 | 88.7 | - | 74.9 | - | - | 2138 | 74.7 | 43.6 | 60.8 | 51.3 | - | - |
| Qwen2.5-VL-3B (Bai et al., 2025) | 3B | 0.6B | 79.3 | 84.0 | 77.1 | 93.9 | - | 81.6 | - | - | 2157 | 79.1 | 53.1 | 61.8 | 62.3 | - | - |
| *Encoder-free Understanding Only Models* | | | | | | | | | | | | | | | | | |
| Mono-InternVL (Luo et al., 2025) | 1.8B | / | 72.6 | 73.7 | 43.0 | 83.0 | 59.5 | 68.6 | 66.7 | - | 1875 | 65.5 | 33.7 | 40.1 | 45.7 | 67.4 | 53.2 |
| EvEv2.0 (Diao et al., 2025) | 7B | / | 71.1 | 73.9 | - | - | 62.9 | 74.8 | - | - | 1709 | 66.3 | 39.3 | 45.0 | - | 71.4 | - |
| VoRA (Wang et al., 2025a) | 7B | / | 56.3 | - | - | - | - | 65.6 | - | 1363 | 1674 | 64.2 | 32.2 | 33.7 | - | 64.2 | - |
| SAIL (Lei et al., 2025) | 7B | / | 77.1 | - | - | - | - | 76.7 | - | - | 1719 | 70.1 | - | 46.3 | 57.0 | 72.9 | - |
| HoVLE (Tao et al., 2025) | 2.6B | / | 70.9 | 78.6 | 55.7 | 86.1 | 64.9 | 73.0 | 71.5 | - | 1864 | 71.9 | 33.7 | 44.3 | 46.2 | 70.7 | 55.6 |
| *Unified Models* | | | | | | | | | | | | | | | | | |
| Emu3 (Wang et al., 2024b) | 8B | 0.3B | 64.7 | 68.6 | 43.8 | 76.3 | 60.3 | 70.0 | 64.0 | - | - | 58.5 | 31.6 | 37.2 | - | 68.2 | - |
| Harmon-1.5B (Wu et al., 2025c) | 1.5B | 0.9B | - | - | - | - | 58.9 | - | - | 1155 | 1476 | 65.5 | 38.9 | - | - | 67.1 | - |
| Show-o2-1.5B (Xie et al., 2025b) | 1.5B | 0.5B | - | - | - | - | 60.0 | 69.0 | - | 1450 | - | 67.4 | 37.1 | - | - | 65.6 | - |
| Janus-Pro-1.5B (Chen et al., 2025c) | 1.5B | 0.3B | - | - | - | - | 59.3 | - | - | 1444 | - | 75.5 | 36.3 | 39.8 | - | - | - |
| **OneCAT-1.5B** | 1.5B | / | 67.0 | 76.2 | 56.3 | 87.1 | 60.9 | 72.4 | 70.0 | 1509 | 1893 | 72.4 | 39.0 | 42.4 | 55.6 | 70.9 | 58.0 |
| ILLUME+ (Huang et al., 2025) | 3B | 0.6B | - | 69.9 | 44.1 | 80.8 | - | 74.2 | - | 1414 | - | 80.8 | 44.3 | 40.3 | - | 73.3 | - |
| Janus-Pro-7B (Chen et al., 2025c) | 7B | 0.3B | - | - | - | - | 62.0 | - | - | 1567 | - | 79.2 | 41.0 | 50.0 | - | - | - |
| Tar-7B (Han et al., 2025a) | 7B | 0.4B | - | - | - | - | 61.3 | - | - | 1571 | 1926 | 74.4 | 39.0 | - | - | - | - |
| Show-o2-7B (Xie et al., 2025b) | 7B | 0.5B | - | - | - | - | 63.1 | 78.6 | - | 1620 | - | 79.3 | 48.9 | - | - | 69.8 | - |
| **OneCAT-3B** | 3B | / | 73.9 | 81.2 | 64.8 | 91.2 | 63.1 | 77.8 | 75.3 | 1630 | 2051 | 78.8 | 41.9 | 52.2 | 61.7 | 72.5 | 63.4 |

Table 4: Evaluation on the DPG-Bench (Hu et al., 2024) and GenEval (Ghosh et al., 2023) benchmarks for visual generation. The dagger ($^\dagger$) indicates methods that employ an LLM for prompt rewriting. Best in **bold**, second best is underlined.

| Model | DPG-Bench | | | | GenEval Benchmark | | | | |
|---|---|---|---|---|---|---|---|---|---|
| | Global | Entity | Attribute | Overall↑ | Single Obj. | Counting | Position | Color Attri. | Overall↑ |
| *Generation-only Models* | | | | | | | | | |
| SD3-Medium (Esser et al., 2024) | 87.90 | 91.01 | 88.83 | 84.08 | 0.99 | 0.72 | 0.33 | 0.60 | 0.74 |
| FLUX.1-dev$^\dagger$ (Labs, 2024) | 82.10 | 89.50 | 88.80 | 84.00 | 0.98 | 0.75 | 0.68 | 0.65 | 0.82 |
| *Unified Models* | | | | | | | | | |
| Emu3-8B$^\dagger$ (Wang et al., 2024b) | - | - | - | 81.60 | 0.99 | 0.42 | 0.49 | 0.45 | 0.66 |
| ILLUME+ 3B (Huang et al., 2025) | - | - | - | - | 0.99 | 0.62 | 0.42 | 0.53 | 0.72 |
| Harmon-1.5B (Wu et al., 2025c) | - | - | - | - | 0.99 | 0.66 | 0.74 | 0.48 | 0.76 |
| Show-o2-7B (Xie et al., 2025b) | 89.00 | 91.78 | 89.96 | 86.14 | 1.00 | 0.58 | 0.52 | 0.62 | 0.76 |
| Janus-Pro-7B (Chen et al., 2025c) | 86.90 | 88.90 | 89.40 | 84.19 | 0.99 | 0.59 | 0.79 | 0.66 | 0.80 |
| Mogao-7B$^\dagger$ (Liao et al., 2025a) | 82.37 | 90.03 | 88.26 | 84.33 | 1.00 | 0.83 | 0.84 | 0.80 | 0.89 |
| BLIP3-o-8B$^\dagger$ (Chen et al., 2025a) | - | - | - | 81.60 | - | - | - | - | 0.84 |
| Tar-7B (Han et al., 2025a) | 83.98 | 88.62 | 88.05 | 84.19 | 0.99 | 0.83 | 0.80 | 0.65 | 0.84 |
| UniWorld-V1-20B (Lin et al., 2025a) | - | - | - | - | 0.99 | 0.79 | 0.49 | 0.70 | 0.80 |
| BAGEL-7B (Deng et al., 2025) | - | - | - | - | 0.99 | 0.81 | 0.64 | 0.63 | 0.82 |
| BAGEL-7B$^\dagger$ (Deng et al., 2025) | - | - | - | - | 0.98 | 0.84 | 0.78 | 0.77 | 0.88 |
| **OneCAT-1.5B** | 90.48 | 86.70 | 86.75 | 81.72 | 0.99 | 0.83 | 0.72 | 0.75 | 0.85 |
| **OneCAT-3B** | 85.46 | 90.81 | 89.00 | 84.53 | 1.00 | 0.84 | 0.84 | 0.80 | 0.90 |

Moreover, OneCAT-3B outperforms recent unified MLLMs that rely on external vision encoders or tokenizers—such as Janus-Pro-7B (using SigLIP) and Tar-7B (using SigLip2)—despite activating fewer parameters. Compared to top-tier encoder-based understanding-only models like Qwen2.5-VL-3B, our model exhibits a slight performance gap, which we primarily attribute to the gap in the scale and quality of training data. Specifically, Qwen2.5-VL was trained on 4T tokens, while OneCAT was trained on only 0.5T tokens for understanding. We believe this gap can be bridged in the future by scaling up the pretraining data and incorporating more higher-quality long CoT instruction data.

**Visual Generation.** We evaluate our model on three widely-used visual generation benchmarks: two for text-to-image generation, GenEval (Ghosh et al., 2023) and DPG-Bench (Hu et al., 2024), and one for image editing, ImgEdit (Ye et al., 2025). To ensure a fair comparison, we adhere to the original raw prompts for the GenEval benchmark, unlike some previous works that employ LLM-based prompt rewriting to enhance performance. Tab. 4 and 5 highlights the highly competitive performance of OneCAT across all tasks.

On GenEval, OneCAT-3B achieves a SOTA overall score of 0.90, surpassing all unified models, including BAGEL-7B (0.88 with rewriting) and Mogao-7B (0.89 with rewriting). Notably, OneCAT-3B excels in challenging categories such as Counting (0.84) and Color Attribute (0.80). On DPG-Bench, OneCAT-3B attains a overall score of 84.53, outperforming strong counterparts like Janus-Pro-7B (84.19) and Mogao-7B (84.33). On ImgEdit-Bench (Tab. 5), OneCAT-3B achieves an a top-tier score of 3.43, outperforms many unified models and specialized editing models. It demonstrates strong capabilities in categories requiring precise local and global adjustments, such as Adjust, Extract, and Background.

Table 5: Evaluation on ImgEdit-Bench (Ye et al., 2025). Higher scores are better for all metrics. Among unified models, best in **bold**, second best is underlined.

| Model | Add | Adjust | Extract | Replace | Remove | Background | Style | Hybrid | Action | Overall |
|---|---|---|---|---|---|---|---|---|---|---|
| *Editing-only Models* | | | | | | | | | | |
| AnyEdit (Jiang et al., 2025) | 3.18 | 2.95 | 1.88 | 2.47 | 2.23 | 2.24 | 2.85 | 1.56 | 2.65 | 2.45 |
| UltraEdit (Zhao et al., 2024a) | 3.44 | 2.81 | 2.13 | 2.96 | 1.45 | 2.83 | 3.76 | 1.91 | 2.98 | 2.70 |
| Step1X-Edit (Liu et al., 2025) | 3.88 | 3.14 | 1.76 | 3.40 | 2.41 | 3.16 | 4.63 | 2.64 | 2.52 | 3.06 |
| ICEdit (Zhang et al., 2025) | 3.58 | 3.39 | 1.73 | 3.15 | 2.93 | 3.08 | 3.84 | 2.04 | 3.68 | 3.05 |
| *Unified Models* | | | | | | | | | | |
| OmniGen (Xiao et al., 2025) | 3.47 | 3.04 | 1.71 | 2.94 | 2.43 | 3.21 | 4.19 | 2.24 | 3.38 | 2.96 |
| OmniGen2 (Wu et al., 2025b) | 3.57 | 3.06 | 1.77 | 3.74 | 3.20 | 3.57 | **4.81** | 2.52 | **4.68** | **3.44** |
| BAGEL-7B (Deng et al., 2025) | 3.56 | 3.31 | 1.70 | 3.30 | 2.62 | 3.24 | 4.49 | 2.38 | 4.17 | 3.20 |
| UniWorld-V1-20B (Lin et al., 2025a) | **3.82** | 3.64 | 2.27 | 3.47 | **3.24** | 2.99 | 4.21 | **2.96** | 2.74 | 3.26 |
| **OneCAT-3B** | 3.65 | **3.70** | **2.42** | **3.92** | 3.00 | **3.79** | 4.61 | 2.23 | 3.53 | 3.43 |

Table 6: Efficiency comparison for understanding (**Left**) and generation (**Right**), tested on one NVIDIA H800. **Left:** We report the time to first token (TTFT). The number of input text tokens are fixed to 24. 256* is the number of visual tokens of thumbnail. **Right:** We report total inference time for Text-to-Image (T2I) and Image-Editing.

| Model | Resolution of Input Image | #Tokens Visual | TTFT(s) | Model | Resolution of Output Image | T2I Infer. Time (s) | Edit Infer. Time (s) |
|---|---|---|---|---|---|---|---|
| Qwen2.5-VL-3B | $768 \times 768$ | 731 | 0.135 | BAGEL-7B | $512 \times 512$ | 8.76 | 13.45 |
| OneCAT-3B | $768 \times 768$ | 731 +256* | 0.067 (50%↓) | OneCAT-3B | $512 \times 512$ | 1.40 (84%↓) | 2.03 (84%↓) |
| Qwen2.5-VL-3B | $1792 \times 1792$ | 4098 | 0.583 | BAGEL-7B | $1024 \times 1024$ | 26.29 | 46.44 |
| OneCAT-3B | $1792 \times 1792$ | 4098 +256* | 0.225 (61%↓) | OneCAT-3B | $1024 \times 1024$ | 2.85 (89%↓) | 4.61 (90%↓) |

**Qualitative results.** We present qualitative comparisons for the text-to-image and image-editing tasks in Fig. 11 and 12 in Appendix, respectively. OneCAT-3B exhibits leading instruction-following and world-understanding capabilities. We also present more visual generation and understanding showcases in Appendix F.

**Comparison of Inference Efficiency.** The left section of Tab. 6 compares the inference efficiency of OneCAT and Qwen2.5-VL for multimodal understanding. Benefiting from our pure decoder-only architecture, which removes the external ViT encoder, OneCAT achieves significantly faster prefilling, reducing first-token latency by up to 61% compared to Qwen2.5-VL on high-resolution inputs. As shown in the right section of Tab. 6, OneCAT exhibits a substantial speed advantage over the diffusion-based BAGEL in image generation. When producing a $1024 \times 1024$ image, OneCAT requires only 2.85s for T2I generation and 4.61s for image editing—approximately 10× faster than BAGEL. This improvement stems from two key elements: our multi-scale autoregressive mechanism within the LLM, and a VAE tokenizer-free design for editing that reduces both encoding and decoding overhead.

## 4.3 ABLATION STUDY

### 4.3.1 ABLATION STUDY ON UNDERSTANDING DISTILLATION (*UND. DISTIL.*) STAGE

We ablate each module in the *Und. Distil.* stage using the OneCAT-1.5B. Each experiment uses 8B sampled tokens (∼10M image-text pairs from Stage-1 dataset) to optimize the `Und. FFN`, followed by a simplified SFT using LLaVA-665k (Liu et al., 2023) dataset. **On Different Distillation Strategies.** As shown in Tab. 7 and Fig. 5(a), our proposed distillation approach—distilling all layers' hidden states—significantly improves visual learning efficiency and multimodal performance. In addition, distilling only the last layer's hidden states or logits also brings gains over the baseline, yet remains inferior to our full distillation approach. This highlights the necessity of full-layer distillation for effective visual knowledge transfer from the custom teacher MLLM. Notably, our strategy yields better performance than applying the distillation approaches of EvE and VoRA to our OneCAT-1.5B under the same setting (as shown in Fig. 6), further validating the efficacy of our proposed method.

**On Different Teachers.** We then analyze the effect of different teacher models. As shown in Tab. 8 and Fig. 5(b), using the pretrained Qwen2.5-VL-1.5B as the teacher results in unstable training and lower distillation efficiency due to parameter misalignment between the teacher and student LLMs—especially for the frozen attention and QKV layers—highlighting the importance of our custom teacher design.

Table 7: Effect of different distillation strategies.

| Methods | MMB | MME-S | MMVet | SEED | AI2D | ChartQA | TextVQA | Avg. |
|---|---|---|---|---|---|---|---|---|
| w/o distillation | 42.0 | 1209 | 13.2 | 47.0 | 53.8 | 10.5 | 10.0 | 31.4 |
| distill last layer's logits | 44.8 | 1312 | 15.4 | 53.0 | 55.2 | 11.5 | 10.6 | 33.9 |
| distill last layer's hidden states | 46.0 | 1255 | 15.3 | 54.3 | 55.3 | 11.0 | 10.9 | 33.9 |
| distill all layers' hidden states | 49.4 | 1327 | 15.5 | 56.3 | 54.4 | 11.9 | 11.3 | **35.3** |
| EvE (Diao et al., 2024) | 44.5 | 1276 | 15.0 | 52.8 | 54.0 | 10.6 | 11.2 | 33.4 |
| VoRA (Wang et al., 2025a) | 41.5 | 1234 | 15.1 | 50.6 | 54.2 | 10.4 | 10.5 | 32.3 |

Table 8: Effect of different teachers. **Avg.** denotes the average score in Tab. 7.

| Methods | Avg. |
|---|---|
| w/o distillation | 31.4 |
| distill with our custom teacher | **35.3** |
| distill with Qwen2.5-VL teacher | 33.7 |

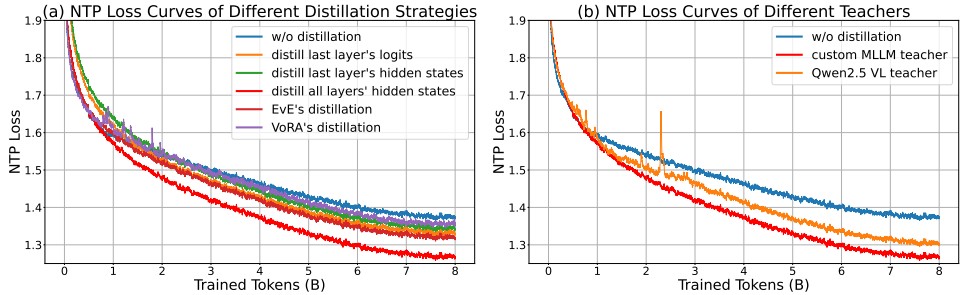

Figure 5: Comparision of different distillation strategies and teachers for stage-1 training.

### 4.3.2 ABLATION STUDY ON UNIFIED MID-TRAINING (STAGE-2)

We initialize the model with OneCAT-1.5B from Stage-1 and perform a simplified unified mid-training to study (i) the effect of token ratio across tasks and (ii) the effect of SAA. All models then undergo a simplified SFT using LLaVA-665k (Liu et al., 2023) dataset for understanding and BLIP3o-60k (Chen et al., 2025a) for generation with a 1:1 token ratio.

**On Token Ratio.** We sample 5B and 10B tokens for text-only (T) and multimodal understanding (U), and vary the number of trained tokens in visual generation (G) to study the impact of token ratio. As shown in Table 9, increasing the training tokens for G does not significantly affect U but improves G. To balance training cost and performance, we adopt a token ratio of 1:2:6 for Stage-2. **On SAA.** We then remove the SAA during the stage-2 training for performance comparison, as shown in Table 10. Each model train 5B, 10B, 30B tokens for T, U, and G for Stage-2, respectively. We can see that removing SAA leads to an obvious performance drop. We also present the visualization of frequency properties of tokens from different scales in Appendix E to better understand the motivation of our SAA.

Table 9: Effect of trained token ratio across text-only (T), understanding (U), and generation (G).

| Token Ratio | Understanding | | | | | | | Generation | |
|---|---|---|---|---|---|---|---|---|---|
| (T:U:G) | MMB | MME-S | MMVet | SEED | AI2D | ChartQA | TextVQA | Avg. | GenEval | DPG |
| 5B : 10B : 0B | 61.6 | 1556 | 28.9 | 67.2 | 60.1 | 56.1 | 56.2 | 55.1 | - | - |
| 5B : 10B : 10B | 62.7 | 1547 | 29.7 | 66.7 | 60.3 | 59.8 | 55.4 | 55.6 | 80.7 | 74.6 |
| 5B : 10B : 30B | 62.0 | 1602 | 29.4 | 66.7 | 59.8 | 61.6 | 55.5 | 56.0 | 81.2 | 74.9 |
| 5B : 10B : 45B | 62.1 | 1544 | 30.1 | 66.8 | 58.4 | 58.6 | 55.0 | 55.2 | 81.4 | 75.6 |

Table 10: Effect of SAA.

| | GenEval | DPG |
|---|---|---|
| w/ SAA | 81.2 | 74.9 |
| w/o SAA | 78.1 | 74.0 |

We present more ablation studies in Appendix D, including (1) the effect of early fusion and late fusion for multimodal understanding, (2) the effect on distilling only visual tokens, and (3) the effect on the increase of training tokens on unified mid-training.

## 5 CONCLUSION

In this work, we presented OneCAT, a pure decoder-only unified multimodal model that seamlessly integrates understanding, generation, and editing within a single, streamlined architecture. By eliminating external encoders and tokenizers, employing a modality-specific MoE design, and introducing a multi-scale autoregressive generation mechanism, OneCAT achieves strong performance across a wide range of benchmarks while significantly improving inference efficiency. Our results demonstrate the viability and advantages of a first-principles approach to multimodal modeling, offering a powerful new baseline for future research and applications in general-purpose multimodal intelligence.

## Ethics Statement

This work presents OneCAT, a unified model for multimodal understanding and generation. We acknowledge the ethical considerations inherent in such technology. The training datasets comprise publicly available and synthetically generated image-text pairs, which may contain societal biases that the model could inadvertently amplify. Potential misuse for creating misleading content is a significant concern. The computational resources required for training such models also entail a non-negligible environmental footprint. These considerations have informed our research process, and we emphasize the importance of developing and evaluating such technologies with safeguards and responsibility.

## Reproducibility Statement

To facilitate the reproducibility of our work, we have provided comprehensive details of the model architecture (Sec. 2), training pipeline (Sec. 3), hyperparameters (Tab. 1), data source (Appendix. H). The code for the model, training procedures, and in-house data, along with instructions for replicating the main results, will be made publicly available. The multi-scale VAE tokenizer is based on the open-source Infinity model (Han et al., 2025b).

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

# Appendix: OneCat: Decoder-Only Auto-Regressive Model for Unified Understanding and Generation

## Contents

## A  RELATED WORK

### A.1  COMPOSITIONAL MLLMs

The field of Multimodal Large Language Models (MLLMs) has rapidly evolved, converging on a dominant **compositional architecture**. This paradigm connects a pre-trained vision encoder (*i.e.,* CLIP (Radford et al., 2021), SigLIP (Zhai et al., 2023), and Intern-ViT (Chen et al., 2024e)), to a powerful LLM through a trainable connector. Pioneering works (Alayrac et al., 2022; Li et al., 2023b) propose sophisticated connector designs. For example, Flamingo (Alayrac et al., 2022) introduces gated cross-attention layers to inject visual information into an LLM, while BLIP-2 (Li et al., 2023b) develops the Q-Former to bridge the modality gap between the vision encoder and LLM. A significant shift occurs with LLaVA (Liu et al., 2023), which simplifies the connector to a lightweight MLP, which become a foundational blueprint for subsequent MLLMs. For example, recent state-of-the-art models like the InternVL series (Chen et al., 2024e;c;d; Zhu et al., 2025; Wang et al., 2025b) and the Qwen-VL series (Wang et al., 2024a; Bai et al., 2025; 2023) adopt this core principle and achieve superior performance by leveraging larger-scale training data and more powerful vision and language foundation models. However, this successful compositional design has inherent drawbacks. The separate nature of the vision and language components complicates the end-to-end optimization process and introduces two critical bottlenecks. First, the sequential nature of the architecture, where the vision encoder must fully process an image before the LLM can begin its generation, leads to high inference latency, especially for the **prefilling** stage. Second, the connector acts as an information bottleneck. In this so-called **late fusion** pipeline, complex visual information is compressed into a compact representation for the LLM, inevitably causing a loss of fine-grained visual detail. These fundamental limitations are now motivating a shift in the field towards more deeply integrated, such as decoder-only models, that aim to overcome these challenges.

### A.2  DECODER-ONLY MLLMs

Decoder-only MLLMs, also known as monolithic MLLMs, have recently emerged as a minimalist yet powerful alternative to the compositional MLLMs. This paradigm aims to achieve greater efficiency and a more direct **early fusion** of modalities by removing the separate vision encoder or tokenizer. For example, Fuyu-8B (Bavishi et al., 2023) processes vision patches by feeding them through a simple linear patch embedding layer directly into the LLM, which markedly reduce inference latency. Inspired by this success, subsequent works (Diao et al., 2024; Luo et al., 2025; Diao et al., 2025; Lei et al., 2025; Wang et al., 2025a; Shukor et al., 2025), further advance decode-only MLLMs by targeting their training processes and architectures. EvE (Diao et al., 2024) and VoRA (Wang et al., 2025a) aligns the LLM's hidden states with semantic features from a pre-trained ViT. However, directly using a smaller model (e.g., a ViT with hundreds of millions of parameters) as the teacher to distill knowledge into a significantly larger LLM (with several billion parameters) may restrict the expressive capacity of the LLM. Differently, Mono-InternVL (Luo et al., 2025) and EvEv2.0 (Diao et al., 2025) adopt a Mixture-of-Experts (MoE) framework, introducing a dedicated *visual expert* to handle visual-specific features more effectively. HoLVE (Tao et al., 2025) prepends a causal transformer to the LLM to explicitly convert both visual and textual inputs into a shared space. Despite these promising advancements, the overall training efficiency of decoder-only MLLMs remains a significant challenge. More importantly, the potential for the decoder-only architecture to create unified models that can seamlessly integrate multimodal understanding, generation, and even image editing capabilities remains a largely unexplored research avenue.

### A.3  UNIFIED VISUAL UNDERSTANDING AND GENERATION

Building on the success of MLLMs, the convergence of visual understanding and generation into a unified framework now represents a key research frontier (Wu et al., 2025a; Chen et al., 2025c; Xie et al., 2025a; Zhou et al., 2024; Wang et al., 2024b; Xie et al., 2024; Li et al., 2025; Pan et al., 2025; Chen et al., 2025a; Lin et al., 2025a; Deng et al., 2025; Liao et al., 2025a). Pioneering unified MLLMs such as Chameleon (Team, 2024),Transfusion (Zhou

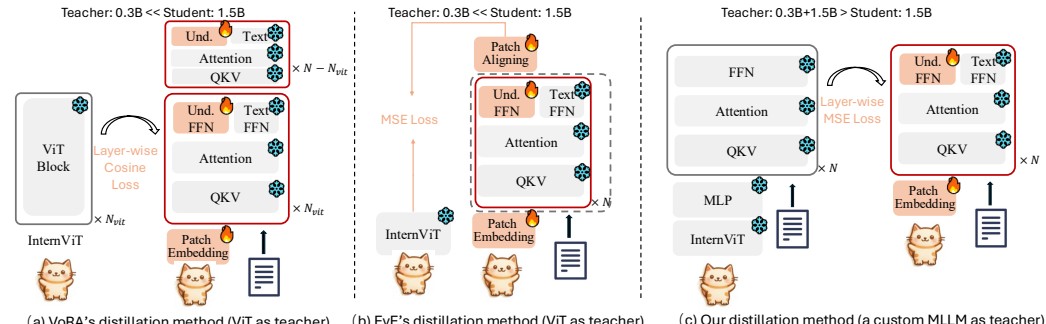

Figure 6: Comparison of existing decoder-only MLLM understanding distillation methods, *i.e.,* VoRA (Wang et al., 2025a), EvE (Diao et al., 2024), when they are applied to our OneCAT. We omit the `Gen. FFN` in this figure for clarity.

et al., 2024), emu3 (Wang et al., 2024b), show-o (Xie et al., 2024) and Synergen-VL (Li et al., 2025) utilize vison tokenizer (e.g., VQ-VAE) to convert images into discrete tokens, enabling seamless multimodal understanding and generation within a single model. However, the discretization inevitably results in lossy visual information and weakens in extracting semantic contents. Janus series (Wu et al., 2025a; Chen et al., 2025c) decouples visual encoding for understanding and generation using two separate encoders, but may compromise performance due to task conflicts in shared LLM parameter space. Metaqueries (Pan et al., 2025), BLIP3-O (Chen et al., 2025a), Uniworld-V (Lin et al., 2025a) assembles off-the-shelf specialized MLMMs and diffusion models by tuning adapters and learnable query tokens, which sacrifices true architectural unification for modularity. BAGEL (Deng et al., 2025) and Mogao (Liao et al., 2025a) employ a Mixture-of-Transformers (MoT) architecture, dedicating different components to autoregressive text generation and diffusion-based visual generation. However, while powerful, this hybrid approach inherits the significant inference latency of diffusion models and still requires separate encoders and tokenizers during the inference.

In contrast to these approaches, our OneCAT introduces a pure decoder-only architecture. By integrating modality-specific experts directly within the decoder, OneCAT achieves versatile multimodal capabilities without the need for external vision encoders or tokenizers at inference time, thus resolving the trade-off between architectural purity and inference efficiency.

## A.4 NEXT SCALE PREDICTION FOR VISUAL GENERATION

Autoregressive models based on next-token prediction(NTP) have long faced efficiency challenges in high-resolution image generation due to the quadratic growth of sequence length with image size. Similarly, diffusion models—though widely successful—often suffer from slow iterative sampling. To address these limitations, VAR (Tian et al., 2024) introduced the next-scale prediction(NSP) paradigm, which encodes images into hierarchical discrete tokens via a multi-scale VAE and generates them autoregressively from low to high resolution, significantly reducing the number of decoding steps. Building upon this, Infinity (Han et al., 2025b) further enhanced this approach with bit-level prediction and extended tokenizer vocabulary, achieving superior generation quality while maintaining efficient inference. To enable unified understanding and generation, VARGPT (Zhuang et al., 2025) stack the transformer from pretrained VAR (Tian et al., 2024) as a visual decoder atop a LLM. However, since the visual tokens (*i.e.,* the input of the visual decoder) **must be decoded token-by-token** through the LLM before subsequent next-scale prediction, this approach compromises the inference efficiency that is the key advantage of the NSP.

In contrast, our proposed OneCAT seamlessly unifies next-token prediction for text generation and next-scale prediction for visual generation **within a single decoder-only**

**transformer of the LLM**, and proposes the scale-aware adapter to further exploit the scale-specific representation for enhanced visual generation.

# B  PRELIMINARY OF NEXT-SCALE PREDICTION

## B.1  MULTISCALE TOKENIZATION

Leveraging the inherent coarse-to-fine structure of natural images, VAR (Tian et al., 2024) introduces a multi-scale tokenizer that encodes an image into $K$ token scales $(R_1, R_2, \ldots, R_K)$. The resolution of each scale $R_k$, denoted as $(h_k, w_k)$, increases monotonically with the scale index $k$. Specifically, given a feature map $F$ extracted from an image with an image encoder, VAR defines these token scales recursively:

$$R_1 = \mathcal{Q}(\text{interpolate}_1(F)), \tag{1}$$

$$R_2 = \mathcal{Q}(\text{interpolate}_2(F - \text{interpolate}_K(R_1))), \tag{2}$$

$$\vdots$$

$$R_k = \mathcal{Q}\Big(\text{interpolate}_k(F - \sum_{i=1}^{k-1} \text{interpolate}_K(R_i))\Big), \tag{3}$$

$$\vdots$$

$$R_K = \mathcal{Q}\Big(F - \sum_{i=1}^{K-1} \text{interpolate}_K(R_i)\Big), \tag{4}$$

where $\text{interpolate}_i$ is an operator that resizes its input to the resolution $(h_i, w_i)$, and $\mathcal{Q}$ is the quantization operator. For a given 3D feature map $x \in \mathbb{R}^{d \times h \times w}$, we implement $\mathcal{Q}$ using Binary Spherical Quantization (BSQ) (Zhao et al., 2024c), following Han et al. (2025b). The quantization is applied to each spatial feature vector $x_{ij} \in \mathbb{R}^d$ as:

$$\mathcal{Q}(x_{ij}) = \frac{1}{\sqrt{d}} \text{sign}\left(\frac{x_{ij}}{\|x_{ij}\|_2}\right). \tag{5}$$

## B.2  VISUAL AUTO-REGRESSIVE TRAINING

The premise for generation is that the feature map $F$ can be well approximated by summing all scales upsampled to the final resolution: $F \approx \sum_{i=1}^{K} \text{interpolate}_K(R_i)$. It therefore suffices to generate the sequence of scales $R_{1:K}$ to synthesize an image. To achieve this, VAR models the joint distribution over the scales auto-regressively, factorizing the log-likelihood as:

$$\log p_\theta(R_{1:K}) = \sum_{k=1}^{K} \log p_\theta(R_k \mid R_{1:k-1}). \tag{6}$$

The model, with parameters $\theta$, is trained to maximize this log-likelihood by learning to predict the current scale $R_k$ conditioned on all preceding scales $R_{1:k-1}$. To enable efficient parallel decoding, VAR assumes that all tokens within the current scale $R_k$ are conditionally independent given $R_{1:k-1}$.

However, this conditional independence assumption, coupled with imperfect model training, can lead to error propagation: mistakes in generating early-stage scales $(R_1, \ldots, R_{k-1})$ are amplified when generating subsequent, higher-resolution scales $R_k$. To mitigate this issue, Han et al. (2025b) proposed *Bitwise Self-Correction*. This technique involves training the model on corrupted versions of the conditioning scales $R_{1:k-1}$, thereby teaching it to generate the correct $R_k$ even when the preceding scales are imperfect. This robustifies the model against its own generation errors during inference.

## B.3  PREDEFINED SCALE SCHEDULES

We follows Han et al. (2025b) and Tian et al. (2024) to establish a set of predefined scale schedules, thus ensuring efficient training across images with varying aspect ratios. As

detailed in Table 11, for each target aspect ratio $r$, we define a specific schedule as a sequence of $K$ resolution tuples: $\{(h_1^r, w_1^r), \ldots, (h_K^r, w_K^r)\}$.

These schedules are designed based on two fundamental principles: **1.Aspect Ratio Consistency:** Each tuple $(h_k^r, w_k^r)$ within a schedule maintains an aspect ratio that is approximately equal to the target ratio $r$, especially at larger scales. **2.Consistent Area Across Scales:** For any given scale level $k$, the image area, calculated as $h_k^r \times w_k^r$, is kept roughly constant across different aspect ratio schedules. This standardization ensures that the training sequence lengths are similar for various aspect ratios, thereby improving overall training efficiency. During the inference stage, these predefined schedules enable the model to generate high-quality images covering a wide range of aspect ratios.

Table 11: Predefined scale schedules $\{(h_1^r, w_1^r), \ldots, (h_K^r, w_K^r)\}$ for different aspect ratios. Following Han et al. (2025b), OneCAT utilizes $K = 13$ scales to generate the highest resolution image, such as a $1024 \times 1024$ image for the 1:1 aspect ratio, while lower-resolution images like $512 \times 512$ can be produced by truncating the schedule to $K = 10$.

| Aspect Ratio | Resolution | Scale Schedule |
|---|---|---|
| 1.000 (1:1) | $1024 \times 1024$ | (1,1) (2,2) (4,4) (6,6) (8,8) (12,12) (16,16) (20,20) (24,24) (32,32) (40,40) (48,48) (64,64) |
| 0.800 (4:5) | $896 \times 1120$ | (1,1) (2,2) (3,3) (4,5) (8,10) (12,15) (16,20) (20,25) (24,30) (28,35) (36,45) (44,55) (56,70) |
| 1.250 (5:4) | $1120 \times 896$ | (1,1) (2,2) (3,3) (5,4) (10,8) (15,12) (20,16) (25,20) (30,24) (35,28) (45,36) (55,44) (70,56) |
| 0.750 (3:4) | $864 \times 1152$ | (1,1) (2,2) (3,4) (6,8) (9,12) (12,16) (15,20) (18,24) (21,28) (27,36) (36,48) (45,60) (54,72) |
| 1.333 (4:3) | $1152 \times 864$ | (1,1) (2,2) (4,3) (8,6) (12,9) (16,12) (20,15) (24,18) (28,21) (36,27) (48,36) (60,45) (72,54) |
| 0.666 (2:3) | $832 \times 1248$ | (1,1) (2,2) (2,3) (4,6) (6,9) (10,15) (14,21) (18,27) (22,33) (26,39) (32,48) (42,63) (52,78) |
| 1.500 (3:2) | $1248 \times 832$ | (1,1) (2,2) (3,2) (6,4) (9,6) (15,10) (21,14) (27,18) (33,22) (39,26) (48,32) (63,42) (78,52) |
| 0.571 (4:7) | $768 \times 1344$ | (1,1) (2,2) (3,3) (4,7) (6,11) (8,14) (12,21) (16,28) (20,35) (24,42) (32,56) (40,70) (48,84) |
| 1.750 (7:4) | $1344 \times 768$ | (1,1) (2,2) (3,3) (7,4) (11,6) (14,8) (21,12) (28,16) (35,20) (42,24) (56,32) (70,40) (84,48) |
| 0.500 (1:2) | $720 \times 1440$ | (1,1) (2,2) (2,4) (3,6) (5,10) (8,16) (11,22) (15,30) (19,38) (23,46) (30,60) (37,74) (45,90) |
| 2.000 (2:1) | $1440 \times 720$ | (1,1) (2,2) (4,2) (6,3) (10,5) (16,8) (22,11) (30,15) (38,19) (46,23) (60,30) (74,37) (90,45) |
| 0.400 (2:5) | $640 \times 1600$ | (1,1) (2,2) (2,5) (4,10) (6,15) (8,20) (10,25) (12,30) (16,40) (20,50) (26,65) (32,80) (40,100) |
| 2.500 (5:2) | $1600 \times 640$ | (1,1) (2,2) (5,2) (10,4) (15,6) (20,8) (25,10) (30,12) (40,16) (50,20) (65,26) (80,32) (100,40) |
| 0.333 (1:3) | $592 \times 1776$ | (1,1) (2,2) (2,6) (3,9) (5,15) (7,21) (9,27) (12,36) (15,45) (18,54) (24,72) (30,90) (37,111) |
| 3.000 (3:1) | $1776 \times 592$ | (1,1) (2,2) (6,2) (9,3) (15,5) (21,7) (27,9) (36,12) (45,15) (54,18) (72,24) (90,30) (111,37) |

## C  Details of Class-Free Guidance

We follow previous works (Chen et al., 2025b; Deng et al., 2025) to use CFG for enhanced visual generation quality. For training, we randomly drop tokens of conditional text and reference image with probabilities 0.1. For inference, we combine conditional and unconditional predicted logits to produce outputs.

Specifically, for **text-to-image generation**, the final logits $\mathbf{L}_{\text{final}}$ are computed as a linear combination of the conditional logits $\mathbf{L}_t$ (with text input) and unconditional logits $\mathbf{L}_\emptyset$ (without text input):

$$\mathbf{L}_{\text{final}} = \lambda_t \cdot \mathbf{L}_t + (1 - \lambda_t) \cdot \mathbf{L}_\emptyset \tag{7}$$

where $\lambda_t$ is the text guidance scale controlling the influence of the text condition.

For **image editing** tasks, which involve both textual and reference image conditions, we employ a dual-guidance mechanism. Let $\mathbf{L}_{t,i}$ denote the logits with both text and reference image conditions, $\mathbf{L}_t$ the logits with text condition only, and $\mathbf{L}_\emptyset$ the logits without any conditions. The refined logits $\mathbf{L}_c$ are first obtained by blending $\mathbf{L}_{t,i}$ and $\mathbf{L}_t$ using a reference image guidance scale $\lambda_i$:

$$\mathbf{L}_c = \frac{\mathbf{L}_{t,i} + \lambda_i \cdot \mathbf{L}_t}{1 + \lambda_i} \tag{8}$$

Then, the final output logits $\mathbf{L}_{\text{final}}$ are computed by combining $\mathbf{L}_c$ with the fully unconditional logits $\mathbf{L}_\emptyset$ using the text guidance scale $\lambda_t$:

$$\mathbf{L}_{\text{final}} = \mathbf{L}_\emptyset + \lambda_t \cdot (\mathbf{L}_c - \mathbf{L}_\emptyset) \tag{9}$$

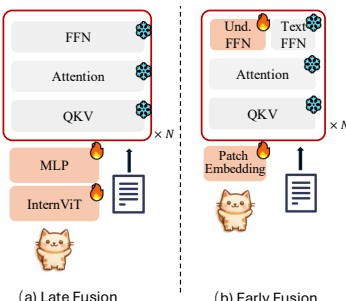

(a) Late Fusion          (b) Early Fusion

Figure 7: Comparison of late fusion and early fusion for multimodal understanding in our setting. We omit the `Gen. FFN` in this figure for clarity.

Table 12: Performance comparison of different fusion strategies and distillation methods for multimodal understanding across varying training scales.

| Methods | #Trained Tokens of Stage-1 | MMB | MME-S | MMVet | SEED | AI2D | ChartQA | TextVQA | Avg. |
|---|---|---|---|---|---|---|---|---|---|
| Encoder-based Late Fusion | 8B | 43.0 | 1222 | 13.3 | 46.8 | 52.1 | 10.1 | 10.6 | 31.4 |
| | 20B | 49.0 | 1437 | 16.7 | 50.4 | 54.4 | 11.6 | 11.3 | 35.0 |
| | 70B | 51.7 | 1426 | 19.2 | 55.4 | 55.9 | 12.0 | 19.7 | 37.8 |
| Decoder-only Early Fusion | 8B | 42.0 | 1209 | 13.2 | 47.0 | 53.8 | 10.5 | 10.0 | 31.4 |
| | 20B | 45.7 | 1312 | 16.7 | 51.8 | 56.4 | 11.7 | 11.9 | 34.4 |
| | 70B | 50.9 | 1423 | 16.9 | 57.4 | 57.2 | 14.2 | 21.0 | 38.3 |
| Decoder-only Early Fusion | 8B | 49.4 | 1327 | 15.5 | 56.3 | 54.4 | 11.9 | 11.3 | 35.3 |
| + Proposed Distillation | 20B | 54.3 | 1410 | 17.7 | 61.0 | 55.4 | 13.6 | 15.8 | 38.3 |
| | 70B | 57.6 | 1476 | 23.4 | 63.0 | 57.2 | 15.0 | 25.0 | 42.0 |
| | 300B | 60.7 | 1526 | 27.1 | 63.4 | 60.0 | 19.2 | 35.7 | 45.8 |

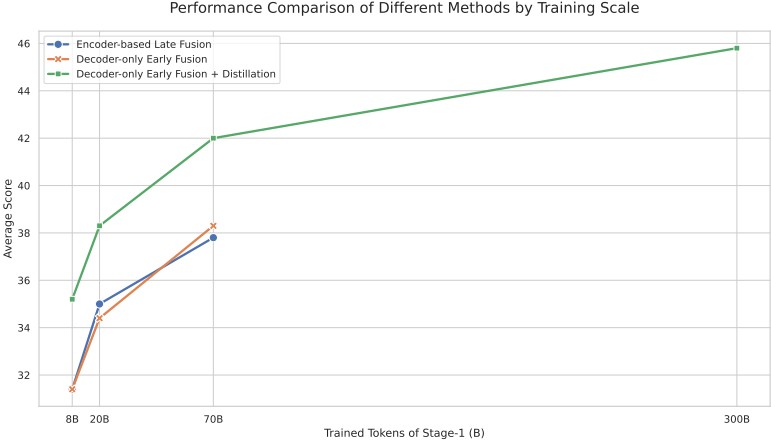

Figure 8: Performance comparison of different methods for multimodal understanding across varying training scales.

This approach allows flexible control over the influence of both textual and visual conditions during the image editing process.

In our experiments, for text-to-image generation, we set $\lambda_t = 20$. For image editing, we set $\lambda_i = 1$ and $\lambda_t = 3$.

## D    Additional ablation study

### D.1    Effect on Early and Late Fusion in MLLM.

We conduct an ablation study to evaluate the impact of encoder-based early fusion versus encoder-free late fusion strategies in multimodal understanding, aiming to validate the effectiveness of the decoder-only architecture employed in our model. As illustrated in Fig. 7, late fusion corresponds to the conventional encoder-based MLLM approach, where images are first processed by a vision encoder before being fed into the LLM. In contrast, early fusion represents the decoder-only MLLM paradigm of our proposed OneCAT.

To ensure a fair comparison, the vision encoder (InternViT) in the late fusion model was randomly initialized without pretrained weights, while the LLM in both models was initialized from the pretrained Qwen2.5-1.5B-instruct and kept frozen, aligning with the setup of our stage-1 training. All variants are further empoloyed a simplified SFT using LLaVA-665k dataset.

Fig. 8 compare the scaling properties of both models with varying trained token budgets for stage-1 and the detailed values are shown in Tab. 12. The experimental results demonstrate that the early and late fusion perform on par, while the early fusion model offers a distinct advantage in computational efficiency, aligning with the findings of Shukor et al. (2025).

### D.2    Effect on Distilling Only Visual Tokens

We use the same setting of Sec. 4.3.2 to conduct an ablation study to evaluate the impact of distilling different types of tokens. Tab. 13 shows that distilling only the continuous visual tokens results in a slight overall performance drop, suggesting that it is crucial to distill both visual and text tokens.

Table 13: Effect of distilling only visual tokens

| Methods | MMB | MME-S | MMVet | SEED | AI2D | ChartQA | TextVQA | Avg. |
|---|---|---|---|---|---|---|---|---|
| w/o distillation | 42.0 | 1209 | 13.2 | 47.0 | 53.8 | 10.5 | 10.0 | 31.4 |
| distill only visual tokens | 48.1 | 1299 | 16.7 | 55.7 | 55.2 | 11.4 | 10.5 | 34.8 |
| distill both visual and text tokens | 49.4 | 1327 | 15.5 | 56.3 | 54.4 | 11.9 | 11.3 | **35.3** |

### D.3    Effect on The Increase of Training Tokens of Unified Mid-Training (Stage-2)

Fig. 9 provides the performance of our OneCAT-1.5B model on multimodal understanding and generation benchmarks at various checkpoints throughout the unified mid-training stage, corresponding to different amounts of training tokens.

Before downstream evaluation, the model of each checkpoint undergoes a simplified unified SFT with a combined instruction dataset (LLaVA-665K for understanding and BLIP3o-60K for generation) , and we oversample the BLIP3o-60K dataset to achieve a 1:1 training token ratio for understanding and generation tasks.

## E    Visualization of discrete visual tokens of different scales

We generate two 1024× 1024 images and present the visualization of discrete visual tokens across different scales and LLM layers. We also visualize the intensity of frequency component by applying Fast Fourier Transform (FFT) to the corresponding tokens' feature maps. As shown in Fig 10, the results show that tokens at lower scales primarily encode low-frequency global information, while higher-scale tokens capture high-frequency details, validating the design rationale of our scale-aware adapter.

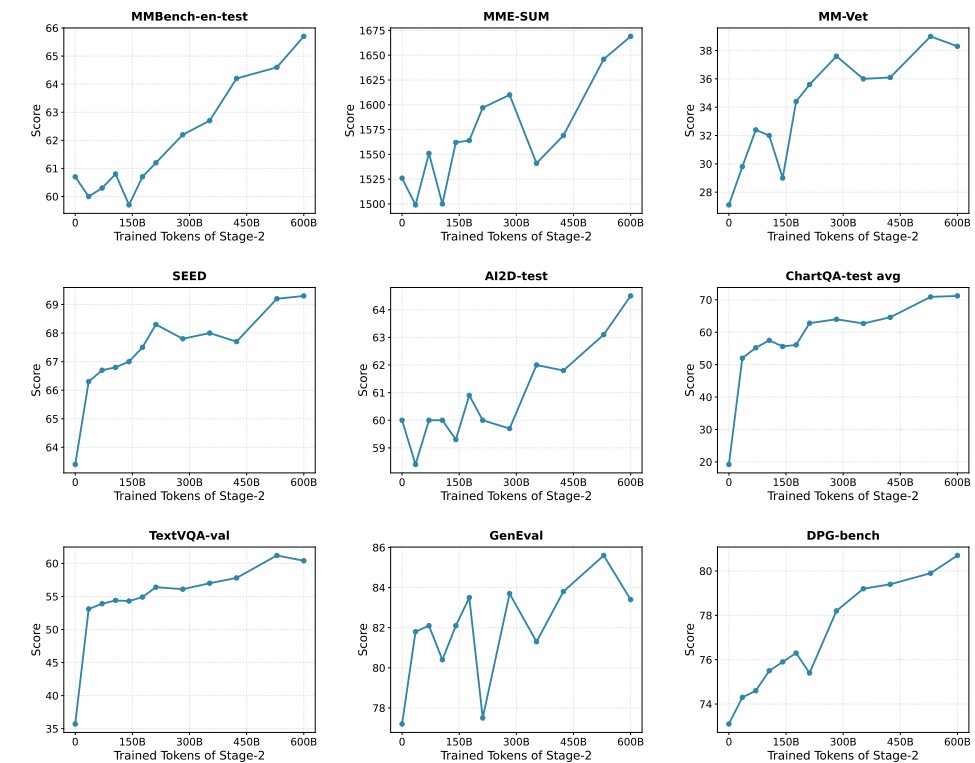

Figure 9: Performance of OneCAT-1.5B on different multimodal understanding and generation benchmarks with the increase of training tokens of unified mid-training (Stage-2).

## F    MORE QUALITATIVE RESULTS

In Fig. 11 and 12, we present qualitative comparisons for text-to-image generation and image editing against several open-source models—including Janus-Pro (Chen et al., 2025c), BAGEL (Deng et al., 2025), and UniWorld-V1 (Lin et al., 2025a)—as well as the proprietary model GPT-4o-image (OpenAI, 2024). We further present additional qualitative results to comprehensively demonstrate the capabilities of our model. Fig. 13 presents text-to-image generation results from OneCAT under various aspect ratios and resolutions. Fig. 14 showcases OneCAT's performance on a range of image editing tasks, such as style transfer, object adjustment, attribute modification, object removal, and background editing. Additionally, Fig. 15 provides examples of OneCAT's multimodal understanding abilities across mathematical reasoning, optical character recognition (OCR), and detailed image captioning.

## G    DETAILED BENCHMARK INFORMATION

In Tab. 3, the following benchmark abbreviations are used: MMB for MMBench-en-test (Liu et al., 2024), MME-P for MME-Perception (Yin et al., 2024), MME-S for MME-Sum (Yin et al., 2024), MMMU for MMMU-Val (Yue et al., 2024), MMVet for MM-Vet (Yu et al., 2023), SEED for Seed-bench (Li et al., 2023a), MathVista for MathVista-testmini (Lu et al., 2023), TextVQA for TextVQA-val (Singh et al., 2019), ChartQA for ChartQA-test (Masry et al., 2022), InfoVQA for InfoVQA-test (Mathew et al., 2022), DocVQA for DocVQA-test (Mathew et al., 2021a), GQA for GQA-testdev (Hudson & Manning, 2019a), and AI2D for AI2D-test (Kembhavi et al., 2016a). For VQA benchmarks, we compute the average scores of TextVQA, ChartQA, InfoVQA, DocVQA, GQA, and AI2D. For general multimodal benchmarks, the average is computed over MME-S (normalized to a 0–100 scale, where a maximum score of 2800 corresponds to 100), MMB, MMMU, MMVet, MathVista, and SEED.

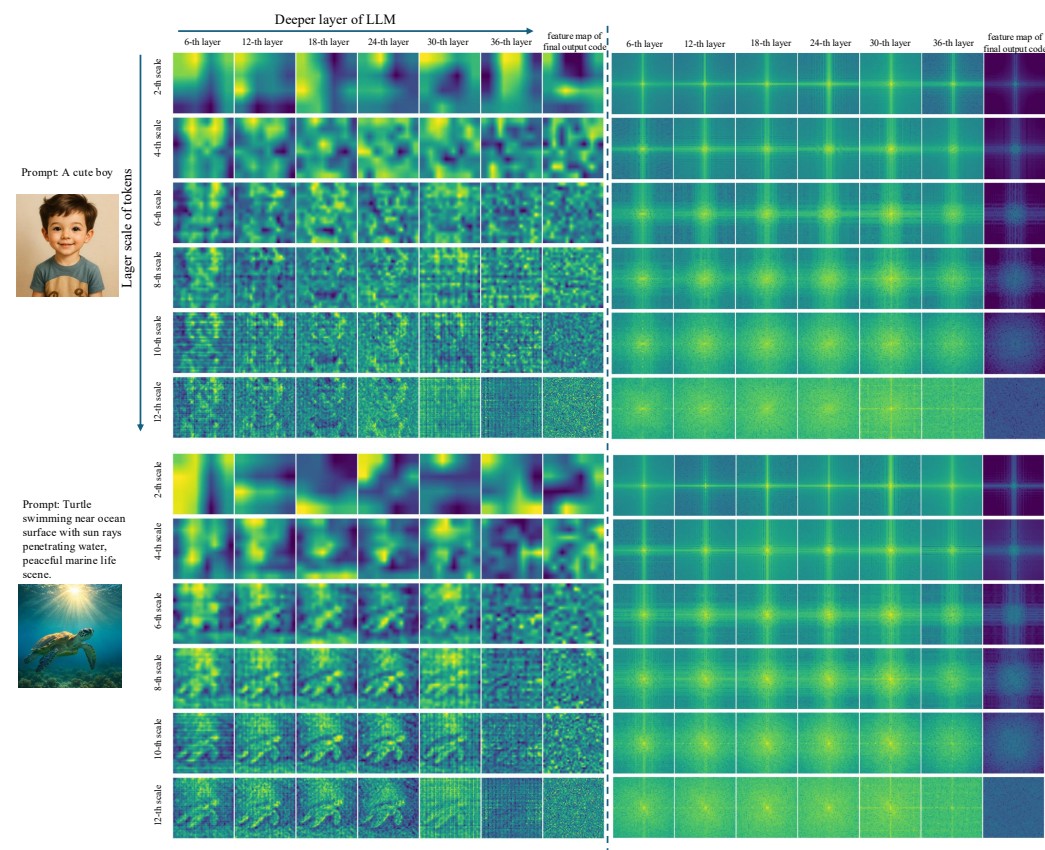

Figure 10: Visualization of discrete visual tokens across scales and LLM layers. **Left:** Each row shows the reshaped feature maps of token hidden states at a specific scale throughout LLM layers. The final column displays the feature map of final output codes fed to the image detokenizer for image reconstruction. All features are resized to 64×64 for display. **Right:** Frequency intensity map of the corresponding feature maps. Lighter colors indicate larger magnitudes, while pixels closer to the center represent lower frequencies. Zoom in better.

## H  DATA SETUP DETAILS

We summary the data source of each training stage in Tab. 14.

**Stage-1:** For the multimodal understanding, we curate a large-scale dataset of approximately 436 million image-text pairs, which is meticulously compiled and processed through comprehensive filtering and deduplication. This dataset is collected from two primary sources: (1) Public Available Image-Text Caption Pairs: We incorporate several publicly available, high-quality image-caption datasets, including Recap-DataComp-1B (Li et al., 2024a), Capsfusion (Yu et al., 2024), Detailed-Caption (Li et al., 2024b), SA1B-Dense-Caption (Data, 2024), and Moondream2-COYO-5M-Captions (isidentical, 2024). (2) Re-captioned Image Datasets: We generate new image-text pairs by re-captioning large-scale public image collections using Qwen2-VL (Wang et al., 2024a). The source image datasets for this process include COYO700M (Byeon et al., 2022), CC12M (Changpinyo et al., 2021), CC3M (Sharma et al., 2018), LAION-400M (Schuhmann et al., 2021), and Zeor250M (Xie et al., 2023). From this large-scale dataset, we randomly sample a small-scale subset of 10 million samples to train the custom teacher.

For image generation, we construct a dataset of 52 million text-to-image samples after a rigorous filtering process to remove samples with low resolution or poor aesthetic scores. This

Table 14: Summary of Datasets Source in Each Training Stage

| Stage | Task Type | Data Sources |
|---|---|---|
| Stage-1 | Multimodal Understanding | Recap-DataComp-1B (Li et al., 2024a), Capsfusion (Yu et al., 2024), Detailed-Caption (Li et al., 2024b), SA1B-Dense-Caption (Data, 2024), Moondream2-COYO-5M-Captions (isidentical, 2024), COYO700M (Byeon et al., 2022), CC12M (Changpinyo et al., 2021), CC3M (Sharma et al., 2018), LAION-400M (Schuhmann et al., 2021), and Zeor250M (Xie et al., 2023). |
| | Visual Generation | ImageNet-1k (Deng et al., 2009), COYO700M (Byeon et al., 2022), LAION-400M (Schuhmann et al., 2021), CC12M (Changpinyo et al., 2021), and additional synthetic images generated by FLUX. |
| Stage-2 | Multimodal Understanding & Text-only Instruction | Detailed-Caption (Li et al., 2024b), ALLaVA (Chen et al., 2024a), ShareGPT4V (Chen et al., 2024b), SA1B-Dense-Caption (Data, 2024), WIT (Srinivasan et al., 2021), pdfa-eng-wds (pdf), UReader (Ye et al., 2023), DVQA (Mathew et al., 2021b), OCR-VQA (Mishra et al., 2019), WebSRC (Chen et al., 2021), GQA (Hudson & Manning, 2019b), visual-genome (Krishna et al., 2017), GRIT (Peng et al., 2023), and additional in-house visual and text-only instruction samples. |
| | Visual Generation | Visual generation data of Stage-1, AnyEdit (Yu et al., 2025), UltraEdit (Zhao et al., 2024b), HQ-Edit (Hui et al., 2024), and OmniEdit (Wei et al., 2024). |
| Stage-3 | Multimodal Understanding & Text-only Instruction | MAmmoTH-VL (Guo et al., 2024), AI2D (Kembhavi et al., 2016b), OKVQA (Marino et al., 2019), VQAv2 (Goyal et al., 2017), ART500K (Mao et al., 2017), ScienceQA (Saikh et al., 2022), GQA (Hudson & Manning, 2019b), CLEVR-Math (Lindström & Abraham, 2022), COCO-ReM (Singh et al., 2024), TallyQA (Acharya et al., 2019), Docmatix (Laurençon et al., 2024), DVQA (Mathew et al., 2021b), DreamSim (Fu et al., 2023), and ShareGPT4o (Chen et al., 2024d). |
| | Visual Generation | UniWorld (Lin et al., 2025b), BLIP3o-60k (Chen et al., 2025a), ShareGPT-4o-Image (Chen et al., 2025b), and additional synthetic data generated by GPT-4o and FLUX using the partial prompts from JourneyDB (Sun et al., 2023). |

collection consists of 1 million class-labeled images from ImageNet-1k (Deng et al., 2009), 20 million pairs from public collections (*i.e.,* COYO700M (Byeon et al., 2022), LAION-400M (Schuhmann et al., 2021) and CC12M (Changpinyo et al., 2021)), and 30 million in-house synthetic images generated by FLUX. The overall training token ratio across multimodal understanding and visual generation samples in Stage-I is approximately **8:1**.

**Stage-2**: In the unified mid-training, for multimodal understanding we leverage an curated dataset of 70 million visual instruction samples. This dataset is specifically curated to be highly diverse tasks, including general VQA, detailed image captioning, OCR, multimodal reasoning(*i.e.,* STEM problem-solving), knowledge, and visual grounding, which are sourced from Detailed-Caption (Li et al., 2024b), ALLaVA (Chen et al., 2024a), ShareGPT4V (Chen et al., 2024b), SA1B-Dense-Caption (Data, 2024), WIT (Srinivasan et al., 2021), pdfa-eng-wds (pdf), UReader (Ye et al., 2023), DVQA (Mathew et al., 2021b), OCR-VQA (Mishra

et al., 2019), WebSRC (Chen et al., 2021), GQA (Hudson & Manning, 2019b), visual-genome (Krishna et al., 2017), GRIT (Peng et al., 2023), and other in-house synthetic visual instruction data.

For visual generation, we supplement the text-to-image samples of Stage-1 with a additional collection of 8 million image editing samples, resulting a total of 60 million visual generation samples. These additional image editing samples are sourced from several public image editing datasets, including AnyEdit (Yu et al., 2025), UltraEdit (Zhao et al., 2024b), HQ-Edit (Hui et al., 2024) and OmniEdit (Wei et al., 2024).

Additionally, we incorporate 40 million text-only instruction samples to preserve the language ability of LLM. To ensure a strong focus on visual generation in Stage-II, we oversample the visual generation data, resulting a final training token ratio of approximately **1 :2 :6** across text-only, multimodal understanding, and visual generation tasks, respectively.

**Stage-3**: In the SFT stage, for multimodal understanding and text-only instruction, we construct a high-quality dataset of 13 million samples. This dataset comprises 10 million filtered samples from MAmmoTH-VL dataset (Guo et al., 2024) and 3 million samples from other open-source datasets AI2D (Kembhavi et al., 2016b), OKVQA (Marino et al., 2019), VQAv2 (Goyal et al., 2017), ART500K (Mao et al., 2017), ScienceQA (Saikh et al., 2022), GQA (Hudson & Manning, 2019b), CLEVR-Math (Lindström & Abraham, 2022), COCO-ReM (Singh et al., 2024), TallyQA (Acharya et al., 2019), Docmatix (Laurençon et al., 2024), DVQA (Mathew et al., 2021b), DreamSim (Fu et al., 2023), ShareGPT4o (Chen et al., 2024d).

For visual generation, we utilize a total of 3 million samples, aggregated from UniWorld (Lin et al., 2025b), BLIP3o-60k (Chen et al., 2025a), ShareGPT-4o-Image (Chen et al., 2025b), and additional synthetic data generated by GPT-4o (Hurst et al., 2024) and FLUX (Labs, 2024) using the partial prompts from JourneyDB (Sun et al., 2023). The overall training token ratio across text-only, multimodal understanding, and visual generation for unified sft is approximately **1 :5 :6**.

# I  OTHER IMPLEMENTATION DETAILS

**Data Packing and Gradient Accumulation:** To optimize workload balance across distributed processes and increase training throughput, we employ a data packing strategy that concatenates multiple variable-length samples into contiguous sequences. Furthermore, to manage the gradient contributions and token ratios between modalities as in Tab. 1, we utilize an *uneven* gradient accumulation strategy: prior to each optimizer step, we accumulate a *distinct* number of micro-batches' gradients for the text and image generation tasks to obtain a gradient of desired token ratios. Such an approach provides fine-grained control over the effective batch sizes of different tasks, ensuring a balanced and stable joint-training.

**Unbiased Global Batch Gradients:** When training on $N$ distributed processes, naively averaging local loss can lead to biased gradients when per-process token counts vary. The ideal objective is to optimize the *Global Batch Loss*, defined as the loss summed over tokens for all micro-batches, normalized by the global token count, denoted as $T_{global}$. To this end, we first prefetch all micro-batches for the next optimizer step, enabling each process to compute the local token counts; a subsequent *All-Reduce* collective operation then aggregates these local token counts into the final global token count, *i.e.*, $T_{global}$. Similar to Liao et al. (2025b), we then employ *Global Batch Reduced Loss* by dividing each micro-batch loss by the averaged token count per process, $\frac{T_{global}}{N}$, which can be shown that the final synchronized gradient for the subsequent optimizer step is mathematically equivalent to the gradient of the global batch loss, enabling training with unbiased gradients.

# J  DECLARATION ON THE USE OF LARGE LANGUAGE MODELS

In this research, large language models (LLMs) are utilized as auxiliary tools to enhance productivity in specific, non-core aspects of the work. Specifically, LLMs are employed to assist with: (1) **Language Polishing**: Improving the fluency and clarity of the writing in

1512
1513
1514
1515
1516
1517
1518
1519
1520
1521
1522
1523
1524
1525
1526
1527
1528
1529
1530
1531
1532
1533
1534
1535
1536
1537
1538
1539
1540
1541
1542
1543
1544
1545
1546
1547
1548
1549
1550
1551
1552
1553
1554
1555

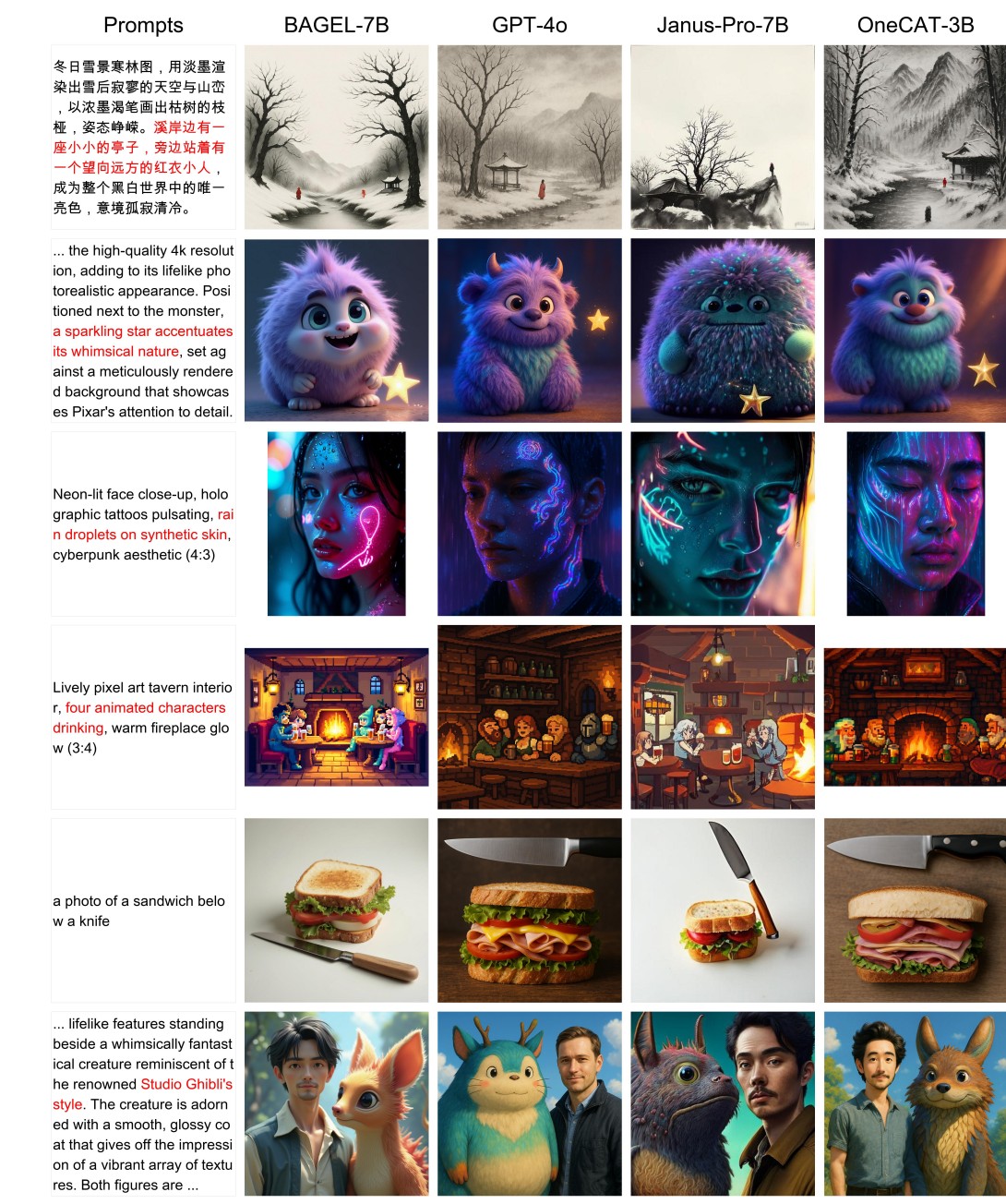

Figure 11: Text-to-Image comparison.

1556
1557
1558
1559

this manuscript. (2) **Code Refactoring:** Aiding in the modification and commenting of code for the project. (3) **Data Curation:** Assisting in the initial filtering and generation of a portion of the training data.

1560
1561
1562
1563
1564
1565

It is important to emphasize that the core scientific contributions of this paper, including the central idea, the architectural design of OneCAT, the novel training methodology, the design and execution of all experiments, and the analysis and interpretation of the results, are conceived and conducted entirely by the human authors. The use of LLMs is strictly limited to execution-level tasks and did not contribute to the intellectual conception or the strategic direction of the research. All outputs generated by LLMs are critically reviewed and edited by the authors, who take full responsibility for the entire content of this work.

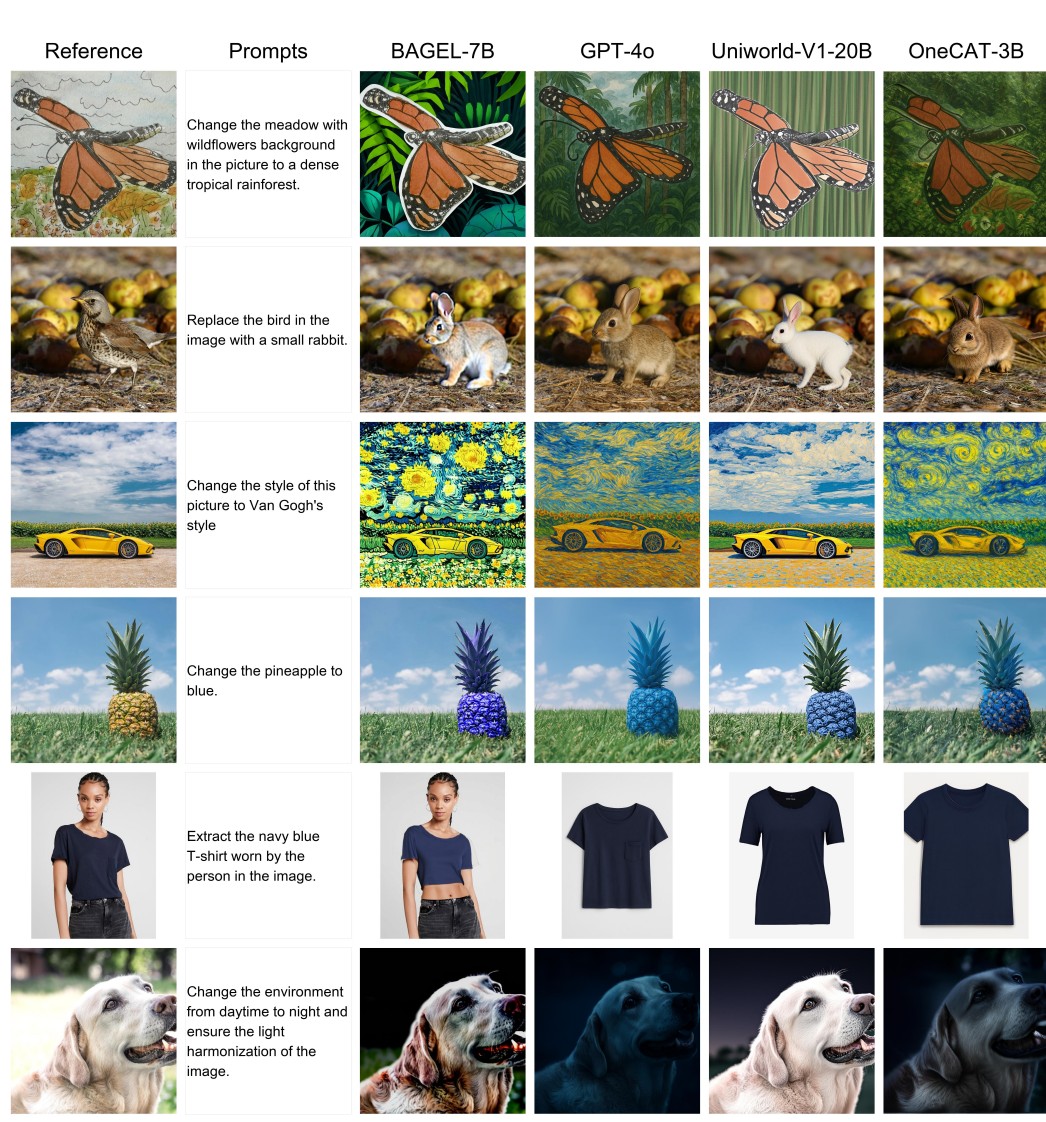

Figure 12: Image-Editing comparison.

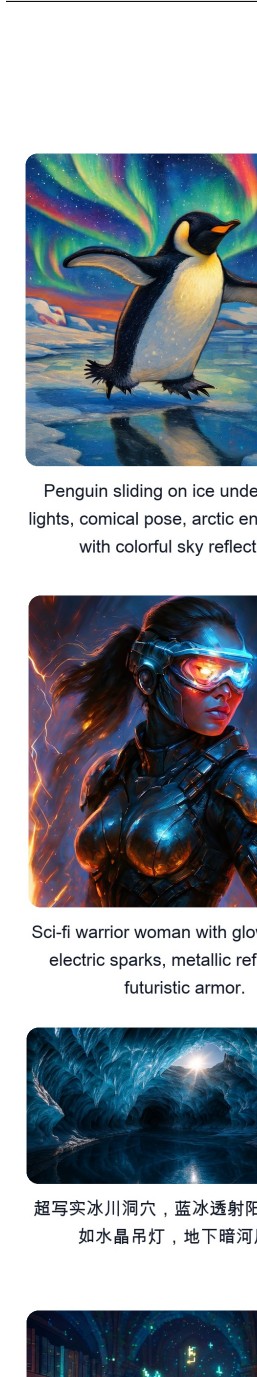

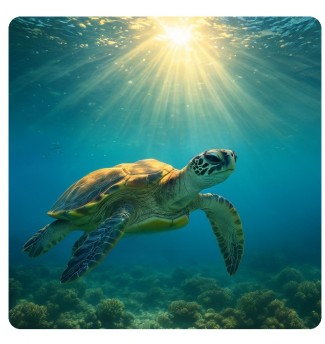

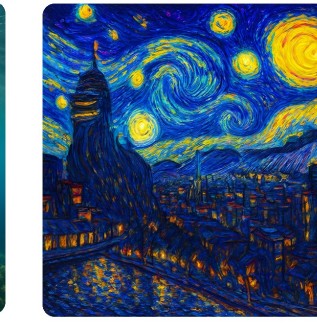

Penguin sliding on ice under aurora lights, comical pose, arctic environment with colorful sky reflection.

Turtle swimming near ocean surface with sun rays penetrating water, peaceful marine life scene.

Van Gogh's Starry Night reimagined with neon cityscape

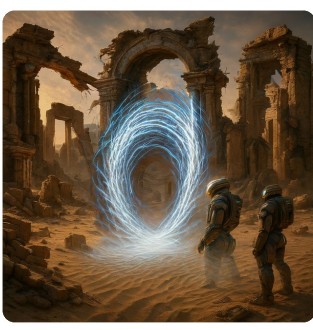

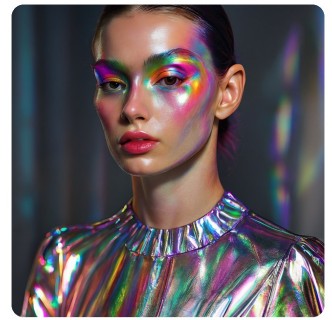

Sci-fi warrior woman with glowing visor, electric sparks, metallic reflections, futuristic armor.

Quantum portal opening in desert ruins, fractal energy waves, archaeologists in exosuits.

Fashion model with iridescent makeup, prismatic light reflections, high-fashion studio setting

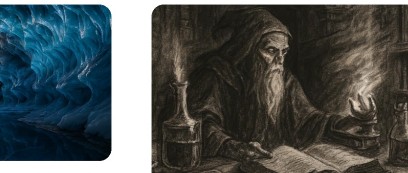

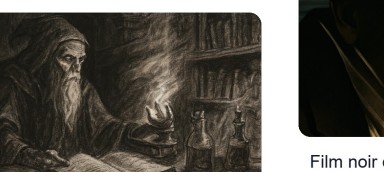

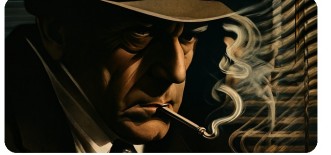

超写实冰川洞穴，蓝冰透射阳光，冰锥如水晶吊灯，地下暗河反光

Charcoal sketch of an old wizard's study, ancient books and potion bottles, dramatic shadows

Film noir detective close-up, venetian blind shadows, cigarette smoke swirls

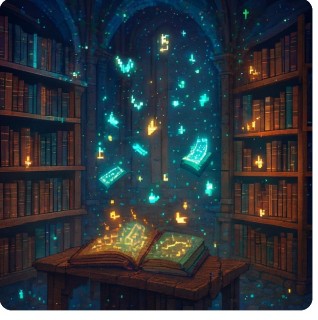

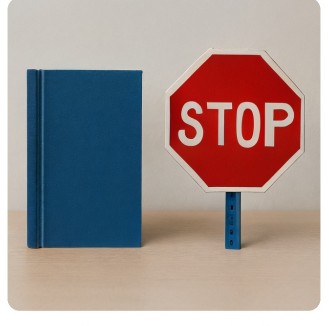

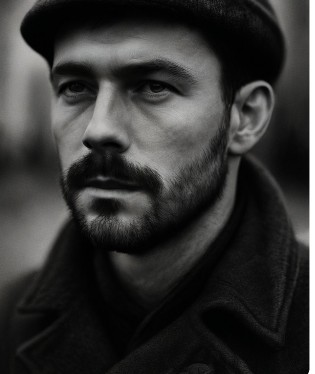

Magical library pixel scene, floating books, glowing runes, enchanted atmosphere.

a photo of a red stop sign right of a blue book

一个戴帽子的男人，特写镜头，黑白照片，高对比度，面部细节清晰，背景模糊，穿着深色外套，胡须和短发

Figure 13: Showcase of the text-to-image abilities of the OneCAT model.

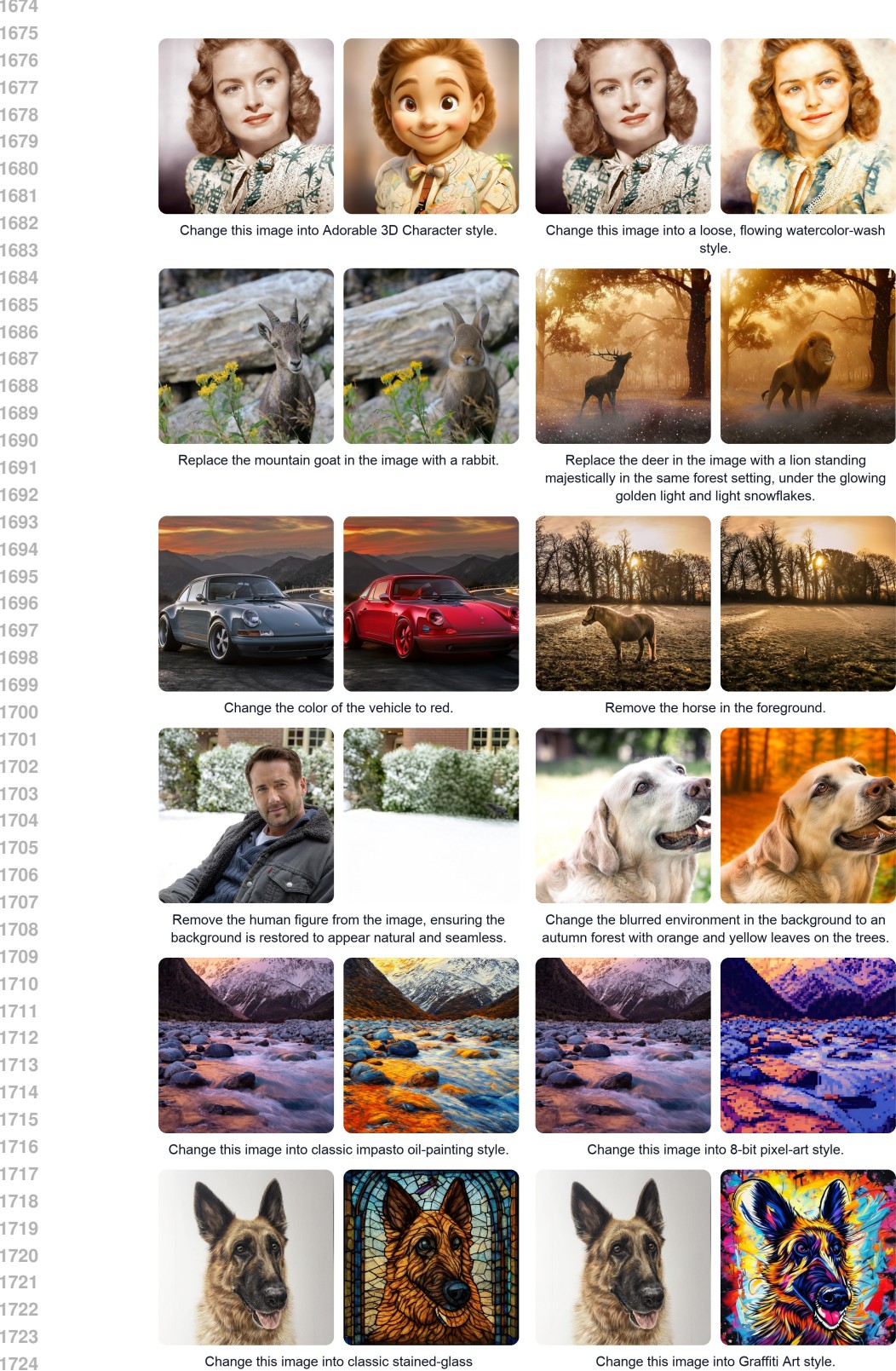

Figure 14: Showcase of the image editing abilities of the OneCAT model.

**Solve the following equations:**

1) $8x + 11 = 4x + 14$

2) $7d - 4 = 11d - 9$

**Question:**
What is d in the last equation?

**Response:**
To find the value of d in the last equation, we need to solve the equation 7d - 4 = 11d - 9 .
Step 1: Subtract 7d from both sides of the equation:
7d - 4 - 7d = 11d - 9 - 7d
-4 = 4d - 9
Step 2: Add 9 to both sides of the equation:
-4 + 9 = 4d - 9 + 9
5 = 4d
Step 3: Divide both sides by 4:
\frac{5}{4} = \frac{4d}{4}
d = \frac{5}{4}
Therefore, the value of d is \frac{5}{4}.

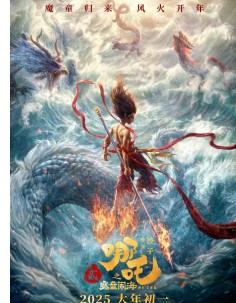

**Question:**
请详细描述该图片。

**Response:**
这是一张电影海报，背景为一片波涛汹涌的海面，海浪翻滚，水花四溅。海面上有几条巨大的龙，它们张牙舞爪，似乎在进行激烈的战斗。海报中央是一个手持长矛的男孩，他站在海浪中，背对观众，面向龙的方向。男孩的头发随风飘扬，身上穿着红色的腰带，腰带上系着红色的丝带。男孩的长矛顶端有火焰，火焰在海浪中闪烁，显得格外醒目。海报上方有一行红色的文字："魔童归来 风火开年"，表示电影的主题和上映时间。海报下方有电影的标题"哪吒之魔童闹海"，以及导演和上映时间的信息。标题下方的"2025 大年初一"表示电影将在2025年大年初一上映。海报的最下方还有两个标志，一个是"可可豆动画"，另一个是"COLD ROOM"。通过这些信息，可以推断出这是一部以"哪吒"为主题的动画电影，讲述的是哪吒在海中大战巨龙的故事。海报通过生动的海浪和龙的描绘，营造出紧张刺激的氛围，吸引观众的注意力。

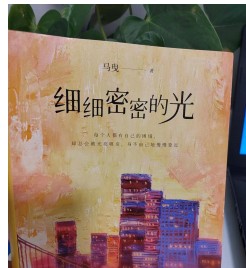

**Question:**
What are all the Chinese scene text in the image?

**Response:**
马曳 著 细细密密的光 每个人都有自己的困境，却总会被光亮吸引，身不由己地慢慢靠近

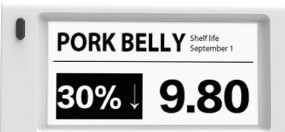

**Question:**
What is the original price for pork belly before discount?

**Response:**
To find the original price for pork belly before the discount, we need to use the given information and perform a simple calculation.
**1. Identify the Discounted Price and Discount Percentage:**
• The discounted price of pork belly is 9.80.
• The discount percentage is 30%.
**2. Understand the Relationship Between Original Price, Discount, and Discounted Price:**
• The discounted price is the original price minus the discount amount.
• The discount amount is calculated as a percentage of the original price.
**3. Calculate the Discount Amount:** Let P be the original price.
• The discount amount is 0.30 \times P (since 30% of P is the discount).
**4. Set Up the Equation:**
• The discounted price is the original price minus the discount amount:[9.80 = P - (0.30 \times P)]
**5. Solve for P:**
• Combine like terms:[9.80 = P - 0.30P][9.80 = 0.70P]
• Divide both sides by 0.70 to isolate P:[P = \frac{9.80}{0.70}][P = 14]
Therefore, the original price for pork belly before the discount is 14.

Figure 15: Showcase of the visual understanding abilities of the OneCAT model.

