# OpenReview forum: "OneCAT: Decoder-Only Auto-Regressive Model for Unified Understanding and Generation"
_ICLR.cc/2026/Conference — ICLR 2026 Conference Withdrawn Submission_

### Official Review · Reviewer_Ymvq · 2025-10-27

**Soundness:** 2
**Presentation:** 3
**Contribution:** 2
**Rating:** 4
**Confidence:** 4

**Summary:**

The paper presents OneCAT, a decoder-only unified multimodal model that removes external vision encoders at inference, routes text and vision tokens to modality-specific FFN experts, and performs multi-scale autoregressive image generation via a Scale-Aware Adapter. Training follows a three-stage recipe that includes teacher-based hidden-state distillation, large-scale mid-training with native and dynamic resolutions, and supervised fine-tuning. The authors report strong understanding scores on common VLM benchmarks, competitive image generation and editing metrics, and latency improvements versus encoder-based understanding and diffusion-based generation systems.

**Strengths:**

The decoder-only formulation is clean and practical for deployment, the modality-specific expert design is straightforward to implement, and the multi-scale adapter provides a concrete mechanism to support autoregressive generation at different scales. The three-stage training pipeline is well organized and the benchmark coverage is broad, with latency tables that highlight potential efficiency gains at inference.

**Weaknesses:**

1. **Overstated novelty and missing isolating ablations:** The core architecture largely reuses a base LLM with duplicated FFNs as modality experts and adds a scale-aware adapter, which reads as an engineering consolidation rather than a new principle. The paper does not isolate the contribution of Modality-MoE versus simpler heads or gating, nor does it compare the adapter to alternative scale-conditioning methods. Adding ablations that replace Modality-MoE with a shared FFN plus a lightweight router and that replace the adapter with per-scale prompts or per-scale layer norms would clarify where the gains originate.

2. **Evaluation fairness and claim calibration:** GenEval results appear to mix raw prompts and LLM-rewritten prompts across systems, and some reported numbers on editing and compositionality are not state-of-the-art when compared side by side. The paper should report like-for-like GenEval in both raw and rewritten settings for OneCAT, mark baselines consistently, include seeds and confidence intervals, and calibrate claims to competitive performance rather than broad SOTA.

3. **Attribution gap due to training-recipe confounds:** The three-stage pipeline introduces strong confounders, yet the paper does not disentangle how much of the improvement comes from the decoder-only architecture (Modality-MoE + SAA) versus the teacher, data scale, or schedule. There is no “no-distillation” control, no reduced-data or reduced-compute runs, no compute-matched comparison to an encoder+adapter baseline, and no evidence that SAA remains necessary without the teacher. As a result, the headline gains cannot be causally attributed to the proposed architecture, which materially weakens the contribution.

**Questions:**

1. **Overstated novelty and missing isolating ablations:** Provide a concise delineation of the technical delta over a shared-FFN baseline and include one small controlled swap where Modality-MoE is replaced by a shared FFN with a lightweight router and SAA is replaced by per-scale prompts or per-scale layer norms under matched settings; if extra runs are infeasible, include an analytic rationale and a few representative failure cases that demonstrate why the proposed choices are necessary.

2. **Evaluation fairness and claim calibration:** State explicitly whether the reported GenEval score uses the LLM-rewriter and whether baselines are aligned; add a short two-row sensitivity table for OneCAT (raw vs. rewritten) with decoding settings, resolution, and seeds, and calibrate any “SOTA” language where competitors lead.

3. **Attribution gap due to training-recipe confounds:** Include a minimal control to isolate architectural gains from recipe effects (e.g., a no-distillation run on a small subset, a compute-matched encoder-plus-adapter baseline, or a teacher-off ablation with SAA), and provide a brief component-wise attribution indicating the approximate fraction of improvement contributed by each element.

---

### Official Review · Reviewer_et3S · 2025-10-28

**Soundness:** 3
**Presentation:** 3
**Contribution:** 2
**Rating:** 4
**Confidence:** 4

**Summary:**

This paper proposes OneCAT, a unified multimodal model that integrates understanding, generation, and editing within a single decoder-only Transformer. It introduces a modality-specific Mixture-of-Experts for efficient multimodal processing and a Scale-Aware Adapter (SAA) to handle hierarchical visual tokens from a multi-scale VAE. The SAA enables dynamic multi-resolution autoregressive generation with reduced latency compared to diffusion-based models. Experiments show that OneCAT achieves competitive or superior results across multimodal benchmarks.

**Strengths:**

1.The paper presents a clean and unified architecture that performs multimodal understanding, generation, and editing within a single decoder-only Transformer, removing external visual encoders or tokenizers and simplifying the overall pipeline.

2.The proposed Scale-Aware Adapter (SAA) effectively handles hierarchical visual tokens, enabling dynamic multi-resolution generation and significantly reducing decoding latency compared to diffusion-based methods.

3.The use of a modality-specific Mixture-of-Experts (MoE) improves computational efficiency by selectively activating experts for different modalities, contributing to strong empirical performance across multiple benchmarks.

**Weaknesses:**

1.The novelty of the work is limited — both the Mixture-of-Experts (MoE) mechanism and the exploration of unified multimodal modeling within a purely autoregressive framework have been widely studied. The contribution here appears more as a system-level integration rather than a conceptual breakthrough.

2. The proposed Scale-Aware Adapter (SAA) seems primarily beneficial for image generation and has little connection to multimodal understanding. Its structure is quite similar to LoRA, and the performance improvement shown in Table 10 appears modest compared to the added computational and architectural complexity. In the unified multimodal setting, the motivation for introducing a module solely aimed at improving generation quality feels somewhat limited.

**Questions:**

see the weakness.

---

### Official Review · Reviewer_5Rua · 2025-10-28

**Soundness:** 4
**Presentation:** 4
**Contribution:** 2
**Rating:** 4
**Confidence:** 4

**Summary:**

This paper introduces **OneCAT**, a unified multimodal model based on a single decoder-only transformer architecture. The model is designed to perform multimodal understanding, generation, and editing tasks seamlessly. The paper demonstrates that a pure autoregressive, decoder-only model can be a sufficient and powerful foundation for general-purpose multimodal intelligence, offering up to 10x faster generation inference than diffusion-based unified models.

**Strengths:**

1. The core contribution is a single decoder-only, autoregressive model that is encoder-free and VAE-tokenizer-free during inference. This elegant design leads to significant, well-documented inference speedups (up to 61% faster prefill, 10x faster generation) by eliminating architectural bottlenecks.
2. OneCAT successfully integrates three distinct modalities of operation (understanding, generation, editing) within a single set of weights, using clever components like the Modality-MoE for task-specific routing.
3. The paper provides robust ablation studies that convincingly justify the design choices, particularly the novel understanding-distillation strategy and the necessity of the custom MLLM teacher.

**Weaknesses:**

1.  The paper candidly notes a performance gap compared to top-tier *encoder-based* understanding-only models (e.g., Qwen2.5-VL-3B). While the authors reasonably attribute this to a significant (8x) difference in training data, it remains a limitation of the current model. It would be valuable to see scaling-law experiments projecting performance with comparable data.
2.  The three-stage training pipeline is highly complex. It involves: 1) training a custom MLLM teacher, 2) a specialized "Expert Pretraining" stage with distillation, 3) unified mid-training, and 4) unified SFT. This complexity, along with the large and diverse data requirements, could be a significant barrier to reproduction and adoption.
3. The ablation study (Table 8) shows that model performance is highly dependent on the *custom* MLLM teacher; using a standard Qwen2.5-VL teacher leads to training instability and worse results. This tight coupling is a potential weakness, as the overall system's success relies heavily on this carefully constructed (and non-trivial) teacher.

**Questions:**

1.  You attribute the understanding performance gap to data scale (0.5T vs 4T tokens). Have you run any scaling experiments (e.g., on smaller models or data subsets) to analyze this trend and project at what data scale OneCAT might close the gap with specialized encoder-based models?
2.  The custom teacher model is critical. The ablation shows it's superior to a standard Qwen2.5-VL teacher. Could you elaborate on *why* this is the case? Is it purely the parameter alignment (as suggested in Sec 4.3.1), or does the teacher's training (MLP-only) result in a different internal representation that is more "distillable" for the decoder-only student?
3.  Figure 10 provides a helpful visualization for the SAA. Beyond this, could you provide any quantitative analysis of specialization? For example, do the different scale-specific adapters show different activation norms or gradient flows when processing tokens from their designated scales vs. other scales?
4.  The CFG scales used seem very different: $\lambda_t=20$ for text-to-image and $\lambda_t=3$ for editing. A scale of 20 is quite high. How sensitive is the model to this hyperparameter? Is there a reason for this large discrepancy between tasks?

---

### Official Review · Reviewer_GN52 · 2025-10-31

**Soundness:** 2
**Presentation:** 2
**Contribution:** 2
**Rating:** 4
**Confidence:** 3

**Summary:**

The paper introduces OneCAT, a single decoder-only multimodal model for image understanding, text-to-image generation, and image editing. It combines a Modality-MoE (text, understanding, generation experts) with multi-scale autoregressive decoding via a Scale-Aware Adapter, and is trained using an encoder teacher with all-layer hidden-state distillation. Experiments show competitive quality across tasks with notably lower inference latency compared to encoder+diffusion pipelines.

**Strengths:**

1. The authors design a teacher tailored to the student’s architecture and distill from all intermediate layers, not just final-layer logits. In an encoder-free unified setup, this is a relatively novel choice and demonstrably improves stability and end-task performance for both understanding and generation.

2. The paper clearly details the training and evaluation pipeline, making the overall methodology easier to follow and assess.

3. The resulting unified model is more efficient than prior baselines, delivering better inference efficiency while maintaining comparable quality.

**Weaknesses:**

1. Limited architectural novelty: The novelty of the model architecture is limited; it only combines previous encoder-free understanding models, such as EVE and VAR. The MoE architecture is also proposed in BAGEL.
2. Unclear objective switch: No clear motivation for replacing the diffusion-like objective used in BAGEL with VAR.
3. Uneven efficiency comparison. Reported speed/latency gains are measured against a larger BAGEL model; results would be more convincing with size-matched baselines or compute-normalized comparisons.
4. “Encoder-free” but still reliant on vision components: Although inference is encoder-free, training depends on (i) a ViT-based teacher for understanding distillation and (ii) a multi-scale VAE tokenizer for generation.
5. Formatting issues. The submission does not follow the official ICLR template, which affects readability.

**Questions:**

1. The custom-trained teacher works better for distillation. Should we interpret this as the capability gap between teacher and student being smaller (and thus easier to match)?

---

### Note · Authors · 2025-11-14

I have read and agree with the venue's withdrawal policy on behalf of myself and my co-authors.